# Dietary Polyphenol Supplementation in Food Producing Animals: Effects on the Quality of Derived Products

**DOI:** 10.3390/ani11020401

**Published:** 2021-02-05

**Authors:** Valentina Serra, Giancarlo Salvatori, Grazia Pastorelli

**Affiliations:** 1Department of Veterinary Medicine, University of Milano, Via dell’Università 6, 26900 Lodi, Italy; 2Department of Medicine and Sciences for Health “V. Tiberio”, University of Molise, Via Francesco De Sanctis 1, 86100 Campobasso, Italy; salvator@unimol.it

**Keywords:** polyphenols, antioxidants, feed additives, food-producing animals, pig, poultry, ruminants, animal product quality

## Abstract

**Simple Summary:**

Polyphenols are secondary plant metabolites mainly known for their antioxidant properties. Their use as feed additives in the nutrition of farm animals is becoming increasingly popular as they are particularly exposed to oxidative stress which is reflected in a lipoperoxidation of the final product. For this reason, it is essential to preserve the quality and the safety of meat and milk products by attenuating oxidative deterioration. Moreover, polyphenols present the advantage of being more acceptable to the consumers than synthetic counterparts, as they are considered to be “non-toxic”. The present review presents an overview of several studies focused on the dietary supplementation of polyphenols to monogastric and ruminants, as well as their direct addition to meat and dairy products, with particular emphasis on their antioxidant effects on the final product.

**Abstract:**

The growing interest in producing healthier animal products with a higher ratio of polyunsaturated to saturated fatty acids, is associated with an increase in lipoperoxidation. For this reason, it is essential to attenuate oxidative deterioration in the derived products. Natural antioxidants such as polyphenols represent a good candidate in this respect. The first part of the review highlights the occurrence, bioavailability, and the role of polyphenols in food-producing animals that, especially in intensive systems, are exposed to stressful situations in which oxidation plays a crucial role. The second part offers an overview of the effects of polyphenols either supplemented to the diet of monogastric and ruminants or added directly to meat and dairy products on the physicochemical and sensorial properties of the product. From this review emerges that polyphenols play an important, though not always clear, role in the quality of meat and meat products, milk and dairy products. It cannot be ruled out that different compounds or amounts of polyphenols may lead to different results. However, the inclusion of agro-industrial by-products rich in polyphenols, in animal feed, represents an innovative and alternative source of antioxidants as well as being useful in reducing environmental and economic impact.

## 1. Introduction

There is a growing interest in producing healthier animal products with a higher ratio of polyunsaturated (PUFA) to saturated fatty acids by modulation of animal’s diet. This nutritional strategy has been associated with an increase in lipoperoxidation. It is essential to preserve the quality and the safety of meat and dairy products by attenuating oxidative deterioration. Antioxidant molecules can be added to feed or introduced directly into the final product to control and reduce the beginning of the oxidative process. Recently, the interest of food processing industries in the use of natural antioxidants rather than synthetic counterparts was increased, for either low environmental impact or economic reasons [1]. Furthermore, natural antioxidants represent a good candidate in this respect because they are also well accepted by the consumers since they are considered safe. In the last decade, there has been a growing interest in supplementing animal feeds with plant antioxidant to boost the nutritional value of meat for consumers’ health benefits [2,3]. For these reasons the calls to use botanical-based feed additives due to their claimed antioxidant activity and beneficial effects on farm animal performance and animal product quality is increasing.

Based on the above, this review aims to briefly summarize the description of polyphenols, and plant or plant extract rich in polyphenols (e.g., rosemary, grape by-products, citrus by-products) that are increasingly used as feed additives in the diet of monogastric and ruminants to mitigate damages caused by oxidative stress to which they are subjected. Moreover, this review offers an overview of the effects of polyphenols when applied in meat and dairy products, discussing the form in which they are used and their influence on the physicochemical and sensorial properties of the final products.

## 2. Classification and Structure of Polyphenols

The polyphenols comprise a large group of more than 8000 different compounds with the phenolic hydroxyl groups being the common structural feature. In nature, polyphenols are usually found conjugated to sugars and organic acids, and, according to the number of aromatic rings and their binding affinity for different compounds, can be divided into three classes, flavonoids, non-flavonoids, and tannins [4] (Figure 1). Flavonoids comprise the largest group of polyphenols with more than 4000 compounds identified and share as a common structure two benzene rings connected by three carbon atoms forming an oxygenated heterocycle [5,6]. Based on the type of heterocycle, the following flavonoid subclasses can be distinguished: flavonols, flavones, flavanols, flavanones, anthocyanins, and isoflavones [7]. The group of flavonoids is responsible for the red, blue, and yellow colouration of plants. They are found mainly in onions, leeks, soybeans, berries, and tea [5,8]. The group of non-flavonoid-polyphenols comprises phenolic acids (cinnamic acid, such as ferulic, caffeic, coumaric, and sinapic acid, and the less abundant hydroxyl benzoic acids, such as gallic and vanillic acid), lignans (e.g., secoisolariciresinol, pinoresinol, syringaresinoland), and stilbenes (e.g., resveratrol). Tannins, commonly referred to as tannic acid, are water-soluble polyphenols that are present in many plant foods. They have been reported to be responsible for decreases in feed intake, growth rate, feed efficiency, net metabolizable energy, and protein digestibility in livestock animals [9].

## 3. The Distribution of Polyphenols in Nature

Polyphenols are commonly found in almost all plants, occurring in various parts of them, such as roots, leaves, flowers, fruit, and seeds, and they protect plants against pests and UV radiation [10,11]. Their distribution at the tissue level is not uniform; for example, the outer layers of plants contain higher polyphenol levels than the inner layers or insoluble polyphenolic compounds are associated with the cell wall, whereas soluble ones are found in the cell vacuoles [12]. Furthermore, the concentration and the proportions of polyphenolic compounds in plants are affected by ripeness at the time of harvest, soil type, sun exposure, air temperature and rainfall, processing, and storage [13]. In particular, the concentration of polyphenols in plants decreases with increasing storage time and in presence of high temperatures due to the high susceptibility of polyphenols to oxidation.

Fruits, vegetables, legume seeds, nuts, herbs, cocoa, and tea represent typical polyphenol-rich sources, and for this reason, humans regularly consume these foods [14,15]. The beneficial effects of polyphenols are mainly attributed to their antioxidant properties as they can act as chain breakers or radical scavengers depending on their chemical structures [16].

### 3.1. Flavonols

Flavonols are the most ubiquitous flavonoids in foods, representing, for example, the most abundant class in *Vitis vinifera* grape berry seeds [17,18]. One of the most important flavonols is quercetin, present in fruit and vegetables, which occurs mainly in leaves and the other parts of the plants as aglycones and glycosides. These compounds are identified by the location of the hydroxyl group on the C ring. The associated sugar moiety is very often glucose or rhamnose, but other sugars may also be involved. Other food sources rich in flavonols are onions, leeks, broccoli, and blueberries.

### 3.2. Flavones

From the structural point of view, flavones lack hydroxyl group at the 3-position. This group of flavonoids derives from two flavanones, naringenin and pinocembrin, that are synthesized from the condensation of one molecule of hydroxycinnamoyl-coenzyme A and three molecules of malonyl-coenzyme A. Flavones can be found in all parts of plants, above and below ground, in vegetative and generative organs, such as stem, leaves, buds, bark, heartwood, thorns, roots, rhizomes, flowers, fruit, seeds, and also in root and leaf exudates or resin [19]. The only important edible sources of flavones identified to date are parsley and celery. 

### 3.3. Flavanols

Flavanols exist in both the monomer form (catechins) and the polymer form (proanthocyanidins). The richest sources of catechins are chocolate and green tea (an infusion of green tea contains up to 200 mg catechins [20]); moreover, but they can be found in many types of fruit and red wine. Black tea contains fewer monomer flavanols, which are oxidized during “fermentation” (heating) of tea leaves to more complex condensed polyphenols known as theaflavins (dimers) and thearubigins (polymers). Catechin and epicatechin are the main flavonols in fruit, whereas gallocatechin, epigallocatechin, and epigallocatechin gallate are found in certain seeds of leguminous plants, in grapes, and more importantly in tea.

### 3.4. Flavanones

This class of flavonoids can occur as O-or C-glycosides and is abundant in citrus fruit and prunes [21]. The hydrolysis of glycoside flavanones produces the not sugar component named aglycones, of which the most common are naringenin in grapefruit, hesperetin in oranges, and eriodictyol in lemons. Flavanones are generally glycosylated by a disaccharide at position seven: either a neohesperidose, which imparts a bitter taste (such as to naringin in grapefruit), or a rutinose, which is flavourless.

### 3.5. Anthocyanins

Anthocyanins represent one of the pigment groups responsible for colour in some fruits and flowers [22,23,24]. They derive from the link of anthocyanidins in one or more glycosidic units, which may be linked to the anthocyanidin by α or β linkage, and always in position 3 of the aglycon. When additional sugars are present in the anthocyanin molecule, they are linked to positions 5 and 7, and less frequently to 3′ and 5′. The sugars encountered in anthocyanins can be hexoses (glucose and galactose) and pentoses (xylose, arabinose) [25]. Anthocyanin molecules are unstable and easily degraded [26] by temperature, pH, light, oxygen, solvents, metallic ions, ascorbic acid, sulphites, and enzymes [22,27]. 

### 3.6. Isoflavones

Isoflavones (family of phytoestrogens) are a group of oxygen heterocycle compounds [28]. The most common isoflavones are daidzein, genistein, and glycitein [29] which occur in legumes, especially in soybean [30]. The main forms of isoflavones in plants are glycosides with sugar such as glucose, malonylglucose, acetylglucose, galactose, and rhamnose.

### 3.7. Phenolic Acids

These compounds consist of a benzene ring bonded to a carboxylic group or a propenoic acid, generating benzoic acids and cinnamic acids respectively. Hydroxybenzoic acids are components of complex structures such as hydrolysable tannins (e.g., gallotannins and ellagitannins). The hydroxycinnamic acids, rarely found in the free form, are more common than hydroxybenzoic acids and consist mainly of p-coumaric, caffeic, ferulic, and sinapic acids [31]. This group of acids is found in all parts of fruit, although the highest concentrations are seen in the outer parts of ripe fruit. Caffeic acid is generally the most abundant phenolic acid and represents between 75% and 100% of the total hydroxycinnamic acid content of most fruit.

### 3.8. Lignans

These phenolic compounds are formed of two phenylpropane units. Lignans are metabolized to enterodiol and enterolactone by the intestinal microflora. The richest dietary source is linseed, which contains secoisolariciresinol and low quantities of matairesinol [32].

### 3.9. Stilbenes

These compounds display two aromatic rings linked by an ethane bridge, and exist in monomeric (resveratrol, oxyresveratrol) and oligomeric form as oligomers of stilbenes (dimmers, trimers, or polymers of resveratrol) or other stilbenes (e.g., epsilon-viniferin, pallidol, etc., [33]). Resveratrol, an important stilbene with anticarcinogenic properties, is found in low quantities in wine and occurs in two isomeric forms, the *trans*- and *cis*-isomers.

### 3.10. Tannins

According to their structure, tannins can be broadly divided into two classes of macromolecules, termed hydrolysable tannins and condensed tannins. Hydrolysable tannins contain a central core of polyhydric alcohol such as glucose, and hydroxyl groups, which are esterified either partially or wholly by gallic acid (gallotannins) or hexahydroxydiphenic acid (ellagitannins [34]). In addition to their astringent character, they have important antioxidant activity [31]. The hydrolysable tannins occur in seed pods, bark and wood, fruits, and leaves or galls of plants belonging to the family Leguminosae, Fabaceae, Combretaceae, and Anacardiaceae. Condensed tannins or proanthocyanidins are high-molecular-weight polymers with a molecular mass of up to 30,000 Da. The monomeric form is a flavan-3-ol (catechin, epicatechin), with a flavan-3, 4-diol, or a leucoanthocyanidin molecule as its precursor. Condensed tannins are widely distributed in fruits, vegetables, forage, plants, cocoa, red wine, and certain food grains, such as sorghum, finger millets, and legume [34]. Table 1 presents polyphenol classes, examples of compounds containing polyphenols, and their presence in plant sources. Figure 2 shows the biological properties of polyphenols found in human, animal models, and livestock, and the main specific effects found in final products are presented in Table 2.

## 4. Bioavailability of Polyphenols in Food-Producing Animals

Bioavailability can be defined as the proportion of bioactive compounds that are successfully absorbed into the bloodstream for metabolic utilization [35]. Polyphenols are considered as bioactive compounds, that unlike macro- and micronutrients, are not essential for life, but have an effect on specific cells and tissues. Their availability is determined by the type of compound, its chemical and physical properties, and the type and presence of functional groups [8,36].

In the monogastric gastrointestinal tract, phenolic compounds metabolism starts from the upper intestinal epithelia and proceeds to the lower intestine, the liver, and then to the peripheral tissues, which include adipose tissue and the kidneys. The route of phenolic absorption can either be via the stomach and small intestine or possibly absorbed by the colon after chemical modification by the colonic microbiota. The microbiota present in the colon allows polyphenols to be absorbed into the bloodstream and subsequently to be excreted either in the urine or via the bile [37]. It has been shown that out of 100% total polyphenolic intake, only 5–10% is absorbed in the small intestine, while the 90–95% will be in the large intestine lumen together with other conjugates excreted by the bile [37,38]. Subsequently, they are exposed to the intestinal enzymes and gut microbiota, which breaks down the polyphenolic structures into smaller molecules to facilitate absorption. In the animal, polyphenols are transformed into glycoside, ester, and polymeric forms that need to be hydrolysed to facilitate absorption by gut microbiota and intestinal enzymes. Phenolic compounds with fewer complex structures are subjected to extensive biotransformation (oxidation, reduction, or hydrolysis) which converts them into water-soluble metabolites in the enterocytes before reaching the liver. The complex phenolic compounds not absorbed in the small intestine reach the colon, where the gut microbiota hydrolyses glycosides resulting in the formation of aglycones. This process decreases the complex structure of the phenolic hydroxyl group into low-molecular-weight phenolic metabolites that can be absorbed. Once absorbed, the molecules reach the liver via the hepatocytes, where they are again subjected to a biotransformation process (conjugation) which improves easy absorption (hydrophobicity) of the molecules and aids in rapid elimination. Finally, the metabolites enter the systemic circulation where they are distributed to the targeted organs or eliminated in the urine [39,40]. A schematic illustration of the described absorption and metabolism of plant polyphenols is summarized in Figure 3.

The content of polyphenols in body tissues is not directly related to their dietary levels [36,42,43]. For example, it has been shown that in pigs whose diets were supplemented with 50 mg/kg quercetin for 4 weeks, quercetin levels were higher in the kidneys (6.31 nmol/g) and colon (13.92 nmol/g) than in the liver (2.83 nmol/g) or plasma (0.67 μmol/L) [44]. On the contrary, when pig diets were supplemented with higher doses of quercetin (500 mg/kg) for 3 days, quercetin levels remained unchanged in bodily tissues (3.78 nmol/g in the liver, 1.84 nmol/g in the kidneys), but increased in plasma (1.1 μmol/L) [45]. In these works, elevated polyphenol concentrations were noted only in organs that participate in the metabolism of the polyphenolic compound. The levels of unabsorbed dietary phenolic compounds exert significant effects on the intestinal environment by suppressing or stimulating the growth of some of the components of intestinal microbiota. The dietary polyphenols present prebiotic properties and exert antimicrobial activities, they enhance the growth of specific beneficial bacteria strains (*Bacillus* spp., *Lactobacillus* spp.) in the intestinal tract while competitively excluding certain pathogenic bacteria and stabilizing gut microbiota; indirectly enhance the host’s immune system and overall health [46,47]. On intestinal morphology, they can exert a positive influence and improve nutrient absorption in monogastric animals [48].

Contrary to what is known about the monogastric, for which it has been clearly established that the small intestine is the main site of absorption for monomeric flavonoids, it is still unknown if flavonoids are absorbed across the rumen epithelium. However, several studies have shown an early peak in plasma flavonoid concentrations after intraruminal administration of quercetin [49] or consumption of a meal containing abundant isoflavones [50], thus suggesting that some flavonoids are adsorbed in the rumen, given that mean ruminal residence time of ingested feed is much greater than 1 h [51]. Consequently, flavonoids that come out of the rumen are probably absorbed in the small intestine due to the increase in the total plasma concentration of flavonol after intraduodenal infusion of quercetin [52]. In dairy cows, it has moreover been shown that the administration of feedstuffs rich in isoflavones (genistein, daidzein) such as soybean meal or red glover silage causes an increase in the concentrations of those isoflavones in blood and milk [53,54]. The supplementation of phenolic compounds in ruminants seems to be effective in preserving polyunsaturated fatty acids (PUFA), rumenic acid, and vaccenic acid from a complete biohydrogenation, with consequent enriching in health-promoting fatty acids in meat and milk, at the expense of saturated fatty acids. The effects of phenolic compounds on cellulolytic bacteria and protozoa probably are also associated with the reduction of both fibre degradability and, indirectly, with methane emission. However, a direct interaction between some specific phenolic compounds (such as hydrolysable tannins) and methanogen microbes was suggested [55].

## 5. Antioxidants and Their Importance in Food-Producing Animals

Τhe efficacy of animal production is based principally on balanced nutritional systems that meet the individual needs of farm animals. Animal growth can be influenced or even regulated by the supplementation of antioxidants in animal feed; thus, it is likely to be affected by the overall redox status of a productive animal [56].

It has been suggested that the high metabolic rate of growing tissues generates abnormally large amounts of free radicals. To avoid the formation of free radicals that usually lead to oxidative stress and other redox-related pathologies, they must be safely scavenged or removed via the administration of antioxidant compounds [57,58]. Oxidative stress is defined as "the imbalance between oxidants and antioxidants in favour of the oxidants, potentially leading to damage" [59]. A particularly destructive aspect of oxidative stress is the excessive production of reactive oxygen species (ROS), such as free radicals and peroxides, that cannot be effectively neutralized by the body [60,61]. In homeostasis, ROS are deactivated by endogenous antioxidants represented by enzymatic antioxidants (superoxide dismutase, catalase, and glutathione peroxidase) and non-enzymatic antioxidants (uric acid, glutathione, coenzyme Q; [61]). In recent years, evidence has emerged that oxidative stress plays a crucial role in the development and perpetuation of inflammation [62], and the potential of plant polyphenols to combat oxidative stress and inflammatory processes in farm animals is fully described by Gessner and colleagues [7]. In pigs, grape seed and grape marc meal extracts or hop extract (10 g/kg diet), have shown to be able to lower the expression of several proinflammatory genes in various portions of the intestine (duodenum, ileum, colon; [63]). The abundance of some potentially pathogenic bacteria was lowered, thus suggesting that polyphenols exerted an antimicrobial effect on pathogenic bacteria in the intestine. The positive effect was reflected in an improvement of the gain-to-feed ratio in pigs, due to an inhibition of pro-inflammatory processes in the intestine and to antimicrobial effects. An improvement in growth performance may be related to the improvement of immune defence and suppression of excessive apoptosis among intestinal epithelial cells, as observed in piglets [64] and poultry [65]. Another action of antioxidants is represented by an improvement of food status and animal feed intake as some plant extracts enhance the flavour and palatability of feed, which improves feed intake and productive performance [66,67]. The increased stimulation of appetite results in higher feed consumption and weight gain. With regards to plant extracts rich in polyphenols, they have been shown to increase digestive secretion such as saliva and digestive enzymes, thus improving the adsorption and utilization of nutrients which increases, in turn, the growth of the animals [68]. Some phenolic compounds like genistein, daidzein, soybean isoflavone, and ferulic acid have been proposed to exert effects on animal metabolism, thus acting as growth promoters by modulating animal metabolism in favour of increasing muscle tissue [69,70] or increasing the bioavailability of nutrients [64]. Studies conducted in ruminants have mainly been focused on inducing changes in the microbial populations of the rumen and its subsequent effects on ruminal fermentation. However, dietary supplementation with ferulic acid to steers has been shown to exert effects similar to β-adrenergic agonists that are used in the final phase of intensive fattening of beef cattle [71].

Moreover, in intensive breeding systems, animals are frequently exposed to oxidation of fatty acids (lipoperoxidation), not only because of their frequent exposure to oxidative stress but also because of their diet [72]. On this basis, numerous studies over the past few years have examined the hypothesis that the supplementation of feed that is enriched with antioxidants to farm animals will provide an improvement of derived product quality [73,74,75,76,77]. As regards meat quality, parameters, such as colour, water-holding capacity, and the oxidative stability of lipids and proteins are usually referred to [77]. Lipid oxidation is a major quality deteriorative process in muscle foods resulting in a variety of breakdown products that produce off-odours and flavours, drip losses, discoloration, loss of nutrient value, decrease in shelf life, and accumulation of toxic compounds [78]. The interaction of alkyl and peroxyl radicals, which are formed during the oxidation of lipids, leads to the formation of non-radical products such as aldehydes [79], directly related to the deterioration of meat colour and flavour, protein stability, and functionality [78].

Another phenomenon closely associated with deteriorative processes that can affect meat and meat products is the oxidation of proteins, which play a fundamental role in meat quality concerning the sensory, nutritional and physico-chemical properties [78]. Protein oxidation occurs through a chain reaction of free radicals like oxidation of lipids in animal muscle. This process induces multiple physico-chemical changes and nutritional value in meat proteins including a decrease in the bioavailability of amino acid protein, change in amino acid composition, decrease in protein solubility due to protein polymerization [80]. Oxidative damage in tissues should be prevented at an early stage by controlling animal diets [37,80,81] and consequently, stabilizing the antioxidant potential of products of animal origin. There is a growing interest in the nutritional aspect of polyphenolic compounds in light of their antioxidant capacity and they may become an important alternative as a partial substitute for vitamin E in animal diets [37]. Polyphenols have a saving action on vitamin antioxidants such as ascorbic acid and tocopherols [82]. Polyphenols, minimizing the adverse consequences of lipid peroxidation by decreasing malondialdehyde concentrations and increasing tocopherol levels in tissues, improve the quality of animal products [4].

## 6. Supplementation of Polyphenols to Animal’s Diet

The use of plant-rich polyphenols seems to be a promising strategy for improving animal product quality through animal diet, and several studies have proven their effects in food-producing animals (Table 2). In fact, the number of publications indexed by Web of Science (https://www.webofknowledge.com (accessed on 23 December 2020)) on polyphenols and their use as feed additives has increased over the years, since 2000 (Figure 4).

The optimum dose of inclusion of polyphenols in animal diets is difficult to define due to the different composition of phenolic compounds present in these products. Due to economic considerations, relevant sources of polyphenols for farm animals are agro-industrial by-products from juice, wine and beer making, such as pomace, peels, seeds, stems and brewery waste, and from the processing of grains, seeds and nuts, such as hulls (rice, buckwheat, almond) and husks (coconut). Polyphenols are present in virtually all matrices of plant origin, but some fruits, more than others, are particularly rich in these bioactive compounds, as in the case of grape. Grape (*Vitis* spp.) represents one of the most consumed and appreciated fruits in the world. Grape pomace, which mainly consists of grape seeds and skins, is the solid by-product of wine-making, approximately representing 20% of the total processed grapes, and has been reported to be a rich source of biologically active compounds, especially polyphenols. Several studies have reported that inclusion of grape by-products in animal diets not only retards oxidation but also improves meat quality when compared to diets with no antioxidants [84,85,94,125,126]. In the last two decades, the use of grape by-products as a dietary supplement for farm animals has been the focus of numerous experimentations. Other natural sources of polyphenols are represented by herbs of the Verbenacee, Lamiaceae family, especially rosemary, sage, oregano and green tea for which a particular emphasis has been given in the nutrition of farm animals. All of these herbs have been reported to possess antioxidant activity.

In the next paragraphs some findings of supplementation of polyphenols to the diet of monogastric and ruminants especially focused on quality parameters of final products, are presented. In fact, it has been demonstrated that dietary polyphenols do not undergo substantial metabolic modifications, thus enriching meat [90,99,112,114,118] and dairy products [107,108,110,115,116,120,121] with bioactive molecules originated from the diet. 

### 6.1. Monogastric

Several studies report the inclusion of grape by-products in poultry and pigs’ diets. Grape represents one of the major sources of phenolic compounds [127]. In this plant, the compounds found in the greatest proportion are the flavanols, which include simple monomers of catechin and its isomer epicatechin, as well as oligomeric proanthocyanidins and polymers commonly known as condensed tannins [128]. Extracts obtained from grape seeds and pomace are complex in composition and contain monomeric phenolic compounds such as (+) -catechins, (−) epicatechin and (−) epicatechin-3-O-gallate, and dimeric, trimeric, and tetrameric proanthocyanidins [129]. It is obvious that the biological activity of these polyphenols depends on their bioavailability. Moreover, not all active compounds have the same availability. For instance, the bioavailability of proanthocyanidins is highly reliant on the extent of polymerization [130], and association with other plant cell constituents like fibre and protein [7]. It has been demonstrated that skin proanthocyanidins have a higher grade of polymerization than proanthocyanidins extracted from seeds [131]. The use of grape by-products as antioxidants in animal feeds has been investigated by various authors not always with consistent results. In broiler chickens, different levels of grape pomace, 30 and 60 g/kg feed from 3 to 6 weeks age [83], or 5 g/kg, 7 g/kg and 10 g/kg for 28 days [86] have been shown to control and decrease thiobarbituric acid reactive substances (TBARS) level by their ability to scavenge free radicals. Although poultry meat contains a low lipid content, its relative concentration of polyunsaturated fatty acids (60% of total fat content) makes it particularly susceptible to oxidative deterioration and the integration results useful. It has been shown that the supplementation of polyphenols improves the antioxidative status and ameliorates oxidative stress in pigs, especially when they are subjected to prooxidative treatment. In fact, it has been reported that the inclusion of 1% of gallic acid, from grapes, wine and tea, with linoleic acid of diets administered for 36 days, improved the nutritional value (higher concentrations of arachidonic acid and docosahexaenoic acid) and water-holding capacity of breast broiler meat [81]. In this paper, the positive results of the treated samples are due to a significantly higher polyphenol content than that of control. Authors reported that gallic acid may directly combine with free radicals and lead to inactivate them which in turn decrease the intracellular concentration of free radicals. The addition of 5% dried grape pomace, a rich source of PUFA, in the diet for finishing pigs, improved the concentration of total n-3 PUFA especially alpha-linolenic fatty acids in *Longissimus dorsi* (LD) muscle after 12 days at 4 °C [132]. Similarly, PUFA content showed an increase in the meat of chicks fed the same grape-product (0, 5, and 10%) [87]. Moreover, it has been reported that polyphenols, due to their 1-electron reduction potentials, may spare vitamin E to delay lipid oxidation and to regenerate tocopherol. In chickens, the dietary supplementation of grape pomace at 8% and grape seed extract at 0.1% determined a significantly higher plasma α- and γ-tocopherol concentration than the control group [133]. Similarly, an increase of α-tocopherol concentration has been observed in the plasma of broiler chickens fed grape pomace at the level of 5 and 10% [134] and in the liver of broilers fed 0.5, 1.5 and 3% [84]. The capacity of grape polyphenols to increase the concentration of vitamin E has also been observed in breast meat of broilers fed a diet supplemented with 100 and 200 mg/kg of a commercial product containing polyphenols derived from grape seeds [135]. This effect results in a better oxidative stability [84]. The administration of feed supplemented with grape pomace to weaned piglets increased significantly n-3 fatty acids such as eicosapentaenoic acid (EPA), docosahexaenoic acid (DHA) and alpha-linolenic acid (ALA) and decreased significantly n-6/n-3 ratio compared with control [136]. The supplementation of grape pomace in laying hens at 2% significantly decreased the egg yolk malondialdehyde (MDA) concentrations in fresh eggs at 0, 30 and 60-min incubations [137]. Moreover, a significantly higher egg weight (*p* < 0.001) has been observed in laying hens fed a diet containing 4% grape pomace compared to the control group, together with a reduction in yolk MDA concentration of either eggs analysed on the first day of collection or eggs stored at +4 °C for 15 days [138]. Grape seeds are considered a relevant source of polyphenol compounds, mainly the monomerics catechin, epicatechin and gallic acid, and the polymeric and oligomeric procyanidins [139]. No influence on antioxidative parameters has been found in pigs receiving a diet supplemented with grape seed extract and bearberry at different levels (100, 300 and 700 mg/kg diet for both extracts; [95]). The supplementation of 5% grape seed to Penedes chickens did not show influences in protein, lipid and ash percentages of meat, but determined a higher percentage of unsaturated fatty acids due to linoleic acid. Concerning sensory analyses of cooked meat from the biceps femoris muscle, the inclusion of grape seed affects some parameters, giving a nuttier smell, more metallic flavour, more stringiness, less pork crackling odour, less pork crackling flavour, less sweet flavour and less broiler meat flavour [140]. These not positive effects found could be due to condensed tannins that can form complexes with proteins which are generally indigestible. Therefore, condensed tannins may have complexed with the protein present in the pelleted pig feed and hence become unavailable for absorption. The supplementation of different levels of grape seed (0.5, 1, 1.5%) and grape seed extract (675, 1350, 2025 mg/kg) was used in laying hens increasing egg production linearly (*p* < 0.01), with the highest egg production value equal to 96.5% in the group receiving 675 mg/kg grape seed extract compared to the control group (77.1%) [141]. Egg quality parameters were not affected by dietary treatments except for albumen index which increased linearly compared to control group, probably due by the protective effect exercised by grape seed on β-ovomucin, directly responsible for the quantity of albumen. 

Another important source of polyphenols is represented by citrus by-products. The genus *Citrus* L. belongs to the family Rutaceae and is one of the most popular fruit crops in the world. It contains active phytochemicals that can protect the health, providing a wide supply of vitamin A, C, and E, folic acid, potassium and pectin. All these phytochemicals possess a wide variety of biological functions including antioxidant, anti-inflammatory, antimutagenic and anticarcinogenic properties [142,143]. In particular, citrus peel, which is most often discarded as waste, contains the highest amount of polymethoxylated flavones compared to other edible parts of the fruit [144]. A study evaluated the effects of including three different levels of ensiled citrus pulp (50, 100 and 150 g/kg diet) in finishing pig diets on carcass quality and other parameters [145]. The results showed an increase of fat deposition at gluteus medius in pigs fed with 50 and 100 g of additive/kg, while the inclusion of 150 g/kg has resulted in a reduction in both energy digestibility and fat deposition. However, the addition of ensiled citrus pulp slightly affects the fatty acid profile of the external sub-cutaneous fat layer, thus resulting benefits in terms of carcass quality. Another study reported that pigs fed citrus pulp (50 and 100 g/kg) had a lower fat thickness at the last rib [146]. Conversely, two works reported that dietary inclusion of 15% dried citrus pulp in diets of finishing pigs did not affect the backfat thickness and meat quality [147,148]. Nevertheless, it has been found that firmness of meat and meat colour were reduced, but juiciness was increased when dried citrus pulp was included at the levels of 22.5% or 30% [148]. In broiler chickens, the dietary supplementation of two different levels of citrus pulp (50 and 100 g/kg) modified the fatty acid composition of breast meat, improving n-3 and n-6 PUFA content, and reduced the breast skin redness, with no influence on meat total lipid content. Additionally, the inclusion of citrus pulp led to a considerable decrease in the levels of vitamin E homologues (α-Tocopherol, γ-Tocopherol, γ-Tocotrienol and δ-Tocopherol; [149]).

Among polyphenols, verbascoside extracted from *Lippia citriodora*, showed antioxidant activity in pork meat and processed product. The effects on carcass characteristics, meat quality, oxidative stability and sensory attributes of *L. dorsi* of dietary supplementation with plant extract from *Lippia* spp., titrated in verbascoside (5 mg/kg feed), have been investigated in pigs from weaning to slaughter (166 days) [150]. Results showed lower (*p* < 0.001) lipid oxidation levels than controls. A reduction (*p* < 0.05) of fat odour and rancid flavour intensity in cooked LD muscle stored at 4 °C for 24 h was observed in the treated group. Even after a short period of dietary supplementation, the same extract was able to significantly (*p* < 0.05) limit the lipid oxidation in fresh muscle and reduce the TBARS concentration in processed pork product like Cremona salami with an improvement in colour that resulted more intensely in the treated group. These studies show that verbascoside is an effective antioxidant in pork meat, enhancing the oxidative status and sensory attributes, without affecting other meat quality parameters. This is in part related to the higher α-tocopherol content in muscle of pigs fed polyphenols, confirming its sparing of Vitamin E. The protective effects of phenolic compounds in the prevention of lipid oxidation have been investigated in other studies. It has been observed that in sausage, apple polyphenols inhibited linoleic acid and cholesterol oxidation through their radical scavenging abilities [151]. Moreover, phenolic compounds inhibit free radical development and the propagation of free radical reactions through the chelation of transition metal ions, principally copper and iron [152]. 

*Rosmarinus officinalis* L. is an aromatic plant originating from the Mediterranean region belonging to the Lamiaceae family. Due to its antioxidant properties of leaves, rosemary has been widely accepted as one of the spices with the highest antioxidant activity, thus replacing or even decreasing synthetic antioxidants in foods [153]. Among the most effective antioxidant constituents of rosemary, the cyclic diterpene diphenols, carnosolic acid and carnosol have been identified. In addition, its extract contains carnosic acid, epirosmanol, rosmanol, methylcarnosate and isorosmanol [154,155]. This plant is a rich source of phenolic compounds and their properties are derived from its extracts [156] and essential oils [157]. The antioxidant properties are correlated to fruiting phases: during the fruiting stage, the increase in the concentration of polyphenols (carnosol, rosmarinic acid and hesperidin), is directly related to the improvement of the extract antioxidant capacity. Hesperidin, significantly influenced the pH and colour and lowered malondialdehyde concentrations in breast muscles of broilers after 3, 6 and 9 days of storage when added to diets in the amount of 3 g/kg of feed [158].

### 6.2. Ruminants

As described above, the effectiveness in the prevention of adverse effects of the oxidative stress has been demonstrated through the supplementation of polyphenols or polyphenols and vitamin E added to compound feeding stuffs for monogastric, while, to the best of our knowledge, less investigated is dietary polyphenols supplementation and the quality of milk and dairy products. The major aim of polyphenols inclusion in ruminant nutrition, linked to the amelioration of food, is the reduction of biohydrogenation of PUFA, and the increase of the accumulation of vaccenic acid (VA) in rumen, increase the amount of these fatty acids reaching mammary gland and, consequently, the amount PUFA, VA and conjugated linoleic acid (CLA; derived from the enzymatic desaturation of VA) in milk. However, literature focusing on the alteration of ruminal biohydrogenation process and manipulation milk fatty acids profile is limited and results are not univocal [159]. Some works reported the ability of polyphenols to decrease SFA and increase the concentration of monounsaturated fatty acids (MUFA) and PUFA in milk [160]. Increased levels of ω-3 FA, linoleic in particular, have been observed in milk from cows fed with a source of tannins [161]. The inclusion of pomegranate extract, as a source of secondary metabolites, in the diet of lactating dairy cows (4.56, 5.58 and 6.60 g of total phenols/kg DM diet), was effective in improving milk fatty acid profile, by decreasing the concentration of SFA and the ration n6/n3 and increasing the amount of EPA and DHA [162].

Anyway, grapes and its by-products have also been used in dairy cows’ diets, for which an interesting perspective is a possibility of enriching milk with substances that have health benefits for its consumers. Dietary supplementation of 15% dried grape pomace to dairy cows in mid-lactation did not influence the milk yield and the milk protein and fat content, but increased the lactose and β-lactoglobulin concentration, giving the milk the properties of a functional food, particularly thanks to the biological activities of β-lactoglobulin, including antiviral action, pathogen adhesion prevention. In a study in which ten Friesian cows received for dietary supplementation of 10% grape pomace (on a dry matter basis) at 56 days, the milk collected at the end of the trial did not show variations in chemical composition [163]. This finding was also confirmed in pasteurized milk cheeses that were analysed after 3, 7, 15 and 30 days from the cheese-making. In that case, slight modifications were observed only for proteolysis and were associated with the action of proteinases and peptidases released by the cheese microbiota. During the cheese manufacturing, milk was pasteurized, with the only microbial forms being represented by *Lactococcus* spp., *Lactobacillus* spp. and *S. thermophilus*, which were used as starters. Such micro-organisms are reported to be responsible for extensive proteolytic activity in cheese, with the consequent production of short peptides and free amino acids. The authors discussed this finding by assuming the role of grape pomace bioactive compounds in favouring the metabolic pathways in lactic acid bacteria, leading to an increased function of proteolytic enzymes. The ability of a grape seed extract in increasing the growth of *Lactobacillus* at the expense of potentially harmful microbial forms, such as *Enterobacteriaceae* and *Clostridium* has been previously reported by another study [164]. 

It has been widely observed that the diet administered to lactating ruminants is commonly responsible for changes in the volatile profile of dairy products, both fresh and ripened [165,166]. It is, therefore, conceivable that compounds present in the diet, or secondary metabolites of the same, can be absorbed by the animal following digestion and then reach the mammary gland and be released into the milk [167]. Following cheese manufacturing, some of these compounds could influence the biochemical mechanisms described above, either directly interacting with the enzymatic forms responsible for these events, or indirectly through the regulation of bacterial gene expression.

Concerning the addition of polyphenols in small ruminant’s diets, it has been observed a negative effect on milk production in Sarda dairy sheep after the dietary inclusion of 75 g of exhausted myrtle berries/day [168]. Depressive effects in the milk yield have been also reported in a study in which sheep were fed diet supplemented with olive leaves [169]. A general depressive effect in milk yield can be observed by increasing the concentration of polyphenols in the diet of sheep and goats, while a positive response in the milk yield is obtained when polyphenols are present at low concentrations [170]. Negative effects on milk composition have also been reported when a dairy sheep diet supplemented with 100 g tomato and grape by-products reduced the protein content caused by the lower rumen degradability of the tomato by-product [168]. Concerning the effects in other ruminants, the addition in lamb of 100 g/kg dry matter of citrus and winery by-products improved the instrumental tenderness of meat by lowering shear force values [171,172]. The same inclusion level in lamb’s diets increased slaughter, warm, and cold carcass weights and longissimus muscle area [132]. Concerning meat sensory attributes, the dietary supplementation of cinnamaldehyde and hesperidin, reported high off-flavour appreciation values in ovine meat [173]. On the contrary, the dietary inclusion of up to 80% dried citrus pulp in the diet of steers, did not negatively influence the texture characteristics and consumer acceptability of beef patties derived from these animals [114]. Goat kids fed with 3.2 mg/day of polyphenols powder extract obtained from Olive mill wastewaters produced meat characterized by a significant reduction of short and saturated fatty acids. Furthermore, a higher (*p* < 0.05) proportion of monounsaturated fatty acids was recorded in the treated group with a concomitant lower (*p* < 0.05) MDA formation in the treatment group compared to control group, and no effects were recorded on PUFA [119]. As reported by authors the acidic composition of meat in ruminants is generally different from that of non-ruminants. The PUFA/SFA ratio is lower because of the hydrogenation of unsaturated fatty acid (UFA) in the rumen, while this process does not occur in monogastric that absorb UFA without any transformation in the gastrointestinal tract. Therefore, several UFA that are normally present in the diet of ruminants, are saturated and cannot be present in derived food. Thus, the diet plays the main role in the modification of fatty acid composition and rumen microbial population, as demonstrated by the evidence that rumen population activity on dietary UFA hydrogenation depends on the administered diet.

## 7. Application of Phenolic Compounds in Food Products 

Plant extracts rich in polyphenols such as herbs and spices, have been used for many years to prevent lipid oxidation, retard development of off flavours and improve colour stability of food [174]. Their increase in the number of studies in food fortification has been mainly due to their antioxidant and free radical-scavenging properties and to their likely positive effects on human health. Moreover, due to their suspected carcinogenic potential, the use of the common synthetic antioxidants such as butylated hydroxytoluene (BHT) and butylated hydroxyanisole (BHA), is cause for concern [175]. Therefore, the use of natural antioxidants has the advantage of being more acceptable to the consumers as they consider these substances to be “non-toxic”. The meat industry is actively looking for solutions increasing product shelf-life. For this purpose, technological strategies involve the application of these natural antioxidants, including the direct application of polyphenols into the meat and meat products or coating of packaging materials and results of several studies are reported (Table 3). Natural antioxidants, in addition to oxidation inhibition, may also affect other products quality attributes, influencing accordingly consumer acceptability (Table 3). 

According to what previously described in animal dietary inclusion, grape by-products are commonly added directly in the final products resulting as an effective antioxidant in poultry, pork, beef and goat meat. The use of 60 mg phenolic content/kg meat of *Niagara* and *Isabel* (*Vitis labrusca*) grape seed and peel extracts was as effective as synthetic antioxidants (0.01% BHT and 0.37% of sodium erythorbate-SE, citric acid, and sugar) at preventing lipid oxidation in raw and cooked chicken meat stored at −18 °C for 9 months [229]. In fact, the control treatment had significantly higher TBARS values (*p* ≤ 0.05) when compared to other treatments with antioxidants. There was no significant difference for oxidation inhibition between the synthetic antioxidants (BHT and SE) and the natural extracts, demonstrating the efficacy of the extracts as an antioxidant in chicken meat. The extract of both varieties did not alter the pH values of raw and cooked samples, confirming studies [230] in which grape seed extract was added to cooked and refrigerated ground chicken meat and cooked and refrigerated chicken breast [176]. Natural extracts used did not alter the colour of raw samples, but especially *Isabel* variety caused a lower intensity of red and yellow colour due to the dark colour of the extract. In the sensory evaluation, no significant alteration in odour and flavour score of the *Isabel*-treated samples has been observed; conversely, the variety *Niagara* interfered with the natural chicken meat flavour and odour, causing a wine/grape odour. Moreover, the grape seed extract reduced attributes associated with warmed-over flavour, such as the musty and rancid odours. The interference in both odour and flavour in the control due to the development of lipid oxidation and consequent development of uncharacteristic flavours has been effectively controlled in both raw and cooked chicken meat with the addition of grape seed extract.

Several studies have reported that antioxidant components in grape by-products can be changed or modified by different extraction procedures. Two different extraction systems (methanolic extraction Type I; methanolic extraction plus High-Low Instantaneous Pressure, Type II) have been used to obtain 0.06 g/100 g grape pomace of final product concentration added in pork burgers packed under aerobic conditions (4 °C) at 0, 3, and 6 days post-storage [231]. The rapid expansion under the controlled condition that occurs in the type II treatment seems to negatively affect the concentration of the phenolic profile. Results in fact indicate the highest anthocyanins content in Type I extract pork burgers and the lowest TBARS during all times of storage. Concerning colourimetric parameters, more intense colour of the raw patties containing the Type I of grape extract has been observed, probably due to the highest content in total anthocyanins which act as potent natural colourants with a positive nutritional and therapeutic effect on human organism derived from its antiradical and antioxidant effect. No negative variation in the redness value has been observed in the burgers containing Type I grape extract, which could be due to the intense antioxidant activity of this compound; it is known that lipid oxidation and myoglobin deterioration are highly correlated, and the secondary products produced in lipid oxidation processes strongly promote colour deterioration.

Concerning beef and beef products, the impact of grape seed extract (GSE) on oxidative stability and shelf-life has been assessed in raw ground beef enriched with omega-3 fatty acids with 250 mg grape seed extract/Kg product during storage at 2 °C for 6 days under retail display conditions [232]. Grape seed extract supplementation had no significant effect on pH during the display; this result confirms previous works conducted on the same natural extract added in ground beef [189,233]. Conversely, grape seed extract has been shown to decrease the oxidation value, since TBARS values of treatments without GSE increased gradually from 0.57 mg MDA/kg to around 3.24 mg MDA/kg during display, whereas treatments with GSE had constant TBARS values. The antioxidant activity of grape seed extract, which is directly linked with the presence of phenolic compounds and can delay the formation of TBARS, has been confirmed once again, in agreement with other studies performed in raw and cooked beef [234,235]. It has been reported that the antioxidant activity of this extract is dependent on the concentration from 0.02% to 0.1% in cooked ground beef [235]. The results obtained suggest that grape seed extract could represent a technologically viable alternative to stabilize lipid oxidation in beef meat as well.

As mentioned above, rosemary extract represents one of the most popular naturally sourced antioxidants. Its use as an antioxidant started in the 1950s [236] and it has been used in the food industry around the world showing the best protection capacity especially in beef products. The effect of rosemary and oregano extracts, added individually (400 mg/kg) or in combination (200 mg/kg), and BHA/BHT, on lipid oxidation and fatty acid composition of irradiated beef burgers stored at −20 °C for 90 days has been investigated [237]. Among the natural additives tested, the highest antioxidant capacity was obtained with rosemary extract. In fact, the results of the study showed that rosemary extract, applied alone and in combination with either BHA/BHT or oregano extract, was more effective in maintaining a low oxidation level in the samples than the oregano extract used individually or in combination with BHA/BHT. The average levels of lipid oxidation of beef burger found in this study during storage time and antioxidant capacity of natural extracts such as rosemary and oregano agree with the results of the previous study in which TBARS values were maintained below 2.0 mg/kg in beef burgers up to 90 days. The results found confirms the great potential to the application of rosemary extract in meat and meat products in replacement of synthetic antioxidants. The combined use of natural extracts and ionizing radiation is recommended to control microbiological and quality changes in beef burger during storage. Interestingly, also in pork products, it has been reported that rosemary extract was more effective than BHA/BHT for preventing increased TBARS values or loss of red colour in frozen pork sausage [238]. Antioxidant activity of 30 mg/100 g rosemary extract has been investigated and compared to BHT, in cooked pork meat patties during 6 days of refrigerated storage at 4 ± 1 °C [239]. The patties supplemented with natural extract showed higher redness values than the control and BHT samples, suggesting that the proportion of oxidized myoglobin is lower. The protective effect of this natural extract against discolouration can be ascribed to the high antioxidant activity of polyphenolic compounds present in rosemary extract, which has been reflected in a total polyphenolic content of 3.9 mg/g in pork patties. Consequently, results showed a significant reduction of TBARS values in batches containing rosemary extract compared to control and BHT samples. These results are in agreement with other studies in which the antioxidant power of rosemary has been demonstrated in meat products such as frozen pork and beef meat [189], refrigerated pork and beef meat [188]. Antioxidant treatment also resulted in lower hexanal content throughout the storage period, showing a higher efficiency against protein oxidation, in comparison with other antioxidants as BHT. Differences in the sensory attributes among batches were not detected by the panel of judges, since aromatic notes that could be attributed to the addition of rosemary extracts caused no negative reactions. The effect of two different levels of rosemary extract (0.3 g/kg and 0.35 g/kg) on storage stability and quality characteristics of ground chicken meat during storage at 4 °C for up to 7 days has been evaluated [240]. The antioxidant effect of rosemary has been compared with the most popular antioxidant used in the meat industry (Ascorbic acid, Sodium nitrite and BHA). Rosemary has demonstrated to be able to delay lipid oxidation and its effect was comparable to the effect of BHA, with TBARS value significantly lower than the control at day 7 of storage time. Similar with the TBARS results, both Sodium nitrite and rosemary extract (0.35 g/kg) showed the highest effect maintaining low total carbonyl values during storage of both cooked and raw meat samples, confirming the ability of rosemary extract to inhibit the formation of aldehydes and protein carbonyls [241,242]. Concerning colour measurements, rosemary extract showed the highest colour stability maintaining redness values compared to the other treatments at day 7 and showed higher lightness values at day 7 compared to the control. Rosemary extract of both levels showed highest significant values of spice odour, but with no ability to distinguish between both levels by the panellists and had a comparable effect on panellist evaluation with Sodium nitrite, which showed the highest overall acceptability. Rosemary at the level of 0.35 g/kg could be an excellent natural antioxidant substitution or partial replacement to the synthetic antioxidant used currently in meat preservation.

Dairy products, which are included in the diet of most of the world’s population, are also enriched with bioactive compounds from natural sources as a result of consumer demands for healthier and functional foods [243]. As well as meat and meat products, one of the most common problems occurring in dairy products rich in polyunsaturated fatty acids is lipid oxidation and the development of off flavours. The supplementation of phenolic compounds in dairy products may affect their sensory and physicochemical properties, in addition to the antioxidant effect. Some protective delivery systems for food application, such as microencapsulation, can be used to avoid degradation and to camouflage the possible unpleasant taste of the added natural compounds, also increasing the product shelf life [244]. The supplementation of free and microencapsulated polyphenols from olive mill wastewater in Greek-type and European-type sheep’s milk yoghurt provided a protective effect that avoided an unwanted drop in the pH during storage [245]. The addition of rosemary extract, accepted as one of the spices with the highest antioxidant activity, proved useful in preventing lipid oxidation also in dairy products [246]. The influence of the incorporation of a free and microencapsulated rosemary extract has been investigated in cottage cheeses [247]. It was observed that free extract affected the moisture, fat, protein and carbohydrate contents, while the microencapsulated forms did not affect the moisture, fat or protein content but increased the ash content and reduced the carbohydrate content significantly. It was also observed that cheeses with microencapsulated extracts preserved their antioxidant characteristics more efficiently during storage. The application of *Inula britannica* on Cheddar-type cheese decreased the pH value and increased the protein and ash contents [248]. In addition, the application of *I. britannica* was related to an increase in the extract taste, bitterness and extract odour ratings but did not have noticeable effects on the extract odour or acid taste. The effects of the application of dried chestnut (*Castanea sativa* Mill.) and lemon balm (*Melissa officinalis*, L.) have been evaluated on Serra da Estrela cheese [249]. The colour measurement revealed that the cheeses enriched with the decoctions of both vegetal samples were slightly darker than the control sample, while the fatty acid profiles showed no significant differences. Higher antioxidant activity was observed in enriched cheeses compared with traditional cheeses. 

## 8. Conclusions

In modern farming systems, the main objective is to obtain products of high quality (milk or meat). The concept of quality does not only include a safe product for the consumer, but also the use of farming practices that respect animals’ health, either in intensive or extensive systems. It should be apparent from this review that polyphenols play an important, though not always clear, role in the quality of meat and meat products, milk and dairy products. To make a comparison among the studies has not been always possible since about 8000 phenolic compounds have been identified and in different amounts in literature. It cannot be ruled out that different compounds and/or different amounts of polyphenols may lead to different results. Anyway, the beneficial effects of plant extracts examined in this review derive mainly from the bioactivities of their polyphenols. The inclusion of these natural extracts in feed rations enhances the oxidative stability of meat and meat products and reduces the number of additives required, like vitamin E and the other synthetic antioxidants, meeting consumer demands for healthier meat products. Similarly, it has been observed that the direct addition of antioxidants into meat and meat products with these plant rich in polyphenols improve their oxidative stability, the overall sensory and nutritional quality of meat and meat products, and thus their shelf life. However, the optimum dose of inclusion of polyphenols in animal diets is difficult to define due to the different composition of phenolic compounds present in these plants. More studies are needed to determine appropriate doses of these plant extracts to induce beneficial effects minimizing negative ones. Finally, the inclusion of agro-industrial by-products in animal feed could also help to reduce the environmental and economic impact associated with their storage and transformation, representing an innovative and alternative source of antioxidants. 

## Figures and Tables

**Figure 1 animals-11-00401-f001:**
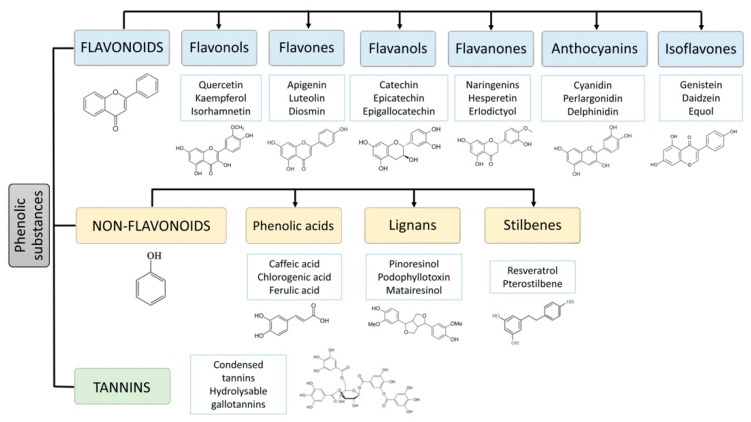
Main classes of polyphenols: flavonoids, non-flavonoids (phenolic acids, lignans, stilbenes) and tannins.

**Figure 2 animals-11-00401-f002:**
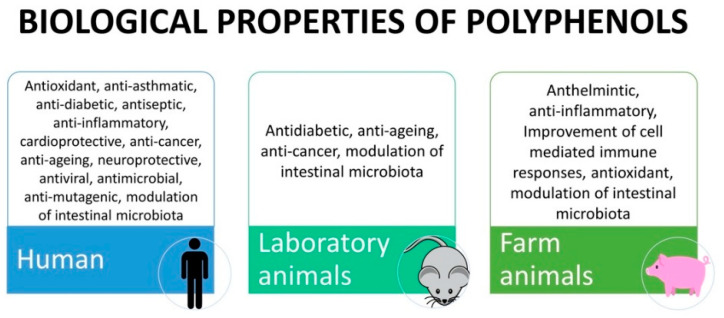
Biological properties of polyphenols exerted in human, animal models and farm animals.

**Figure 3 animals-11-00401-f003:**
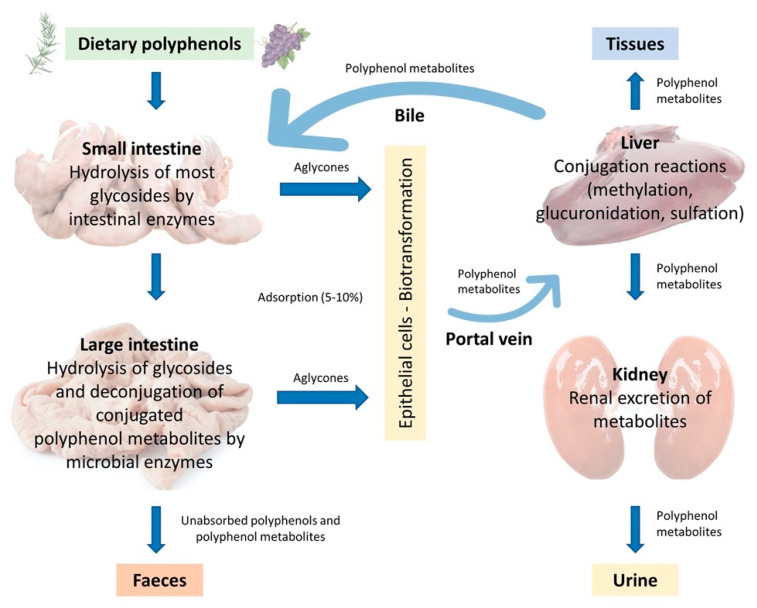
Adsorption and metabolism of plant polyphenols in monogastric farm animals (Adapted from Marín et al., 2014 [41]).

**Figure 4 animals-11-00401-f004:**
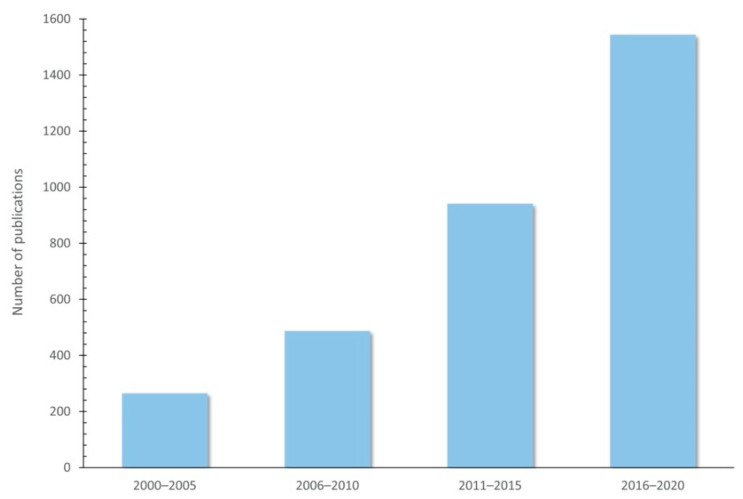
Increasing publications on polyphenols and their use as feed additives since 2000 (Indexed by Web of Science).

**Table 1 animals-11-00401-t001:** Polyphenolic classes, compounds and plant sources.

Group	Class	Compounds	Plant Sources
Flavonoid-polyphenols	Flavonols	Quercetin, kaempferol, isorhamnetin	*Vitis vinifera* grape berry skins, onions, leeks, broccoli, black tea, lettuce, apples, green tea, wine, dill weed
	Flavones	Apigenin, luteolin, diosmin	Celery, red pepper, lemon, onion, oregano, rosemary, parsley, trollflowers
	Flavanols	Catechin, proanthocyanidins, epicatechin, epigallocatechin	Tea, grapes, red wine, apples, blackberries, apricots, dark chocolate
	Flavanones	Naringenins, hesperetin, eriodictyol	Grapefruit, oranges, tangerines, peppermint, lemons, limes, defatted olive
	Anthocyanins	Cyanidin, perlargonidin, delphinidin	Blackberries, cherries, strawberries, raspberries, chokeberries, tomatoes, grapes, green coffee beans, red cabbage, sweet potatoes
	Isoflavones	Genistein, daidzein, equol	Peas, soybean, lentils, red kidney beans
Non-flavonoid-polyphenols	Phenolic acids	Caffeic acid, chlorogenic acid, ferulic acid	Coffee, olive, cabbage, apples, cherries, tomatoes, pears, green coffee beans, dried ginger
	Lignans	Pinoresinol, podophyllotoxin, stegananic, matairesinol	Linseeds, sesame seeds, chives, nuts, roots, leaves, vegetables, spices, cereals
	Stilbenes	Resveratrol, pterostilbene	Almond, chocolate, seeds, and skins of grapes, red wine, peanuts, blueberries, raspberries
Tannins		Tannic acid, Chinese tannin, Turkish tannin, acer tannin, ellagitannin, chebulagic acid	Bean seed coats, persimmons, green coffee beans, mango kernels, pomegranates, strawberries, walnuts, almonds

**Table 2 animals-11-00401-t002:** Natural sources of antioxidants used as feed additives for monogastric and ruminants and main effects found in final products.

Animal Species	Animal Product	Source	Dose Diet	Concentration Product	Storage Days	Effect	Reference
Chicken	Raw and cooked breast meat patties	Grape (*Vitis vinifera*) pomace concentrate	60 g/kg	-	20	Inhibitory effect on lipid oxidation; increase in radical scavenging capacity	[83]
Chicken	Breast and thigh meats	Grape (*V. vinifera*) pomace	30 g/kg	-	7	Inhibitory effect on lipid oxidation (decreased MDA values)	[84]
Chicken	Breast meat	Grape (*V. vinifera*) pomace concentrate	60 g/kg	-	7	Inhibitory effect on lipid oxidation (decreased MDA values); improvement of antioxidant activity (in diet, ileal content, and excreta)	[85]
Chicken	Breast meat	Grape (*V. vinifera*) pomace	10 g/kg	-	10	Inhibitory effect on lipid oxidation; decrease of redness and yellowness values	[86]
Chicken	Thigh meat	Grape (*V. vinifera*) pomace	100 g/kg	-	4	Inhibitory effect on lipid oxidation; increase in the meat PUFA concentration	[87]
Chicken	Breast and thigh meats	Dry rosemary (*Rosmarinus* *officinalis* L.) leaves (R) and rosemary essential oil (RO)	11.5 g/kg (R) 0.2 g/kg (RO)	-	5	Inhibitory effect on lipid oxidation (decreased MDA values) decrease in pH value, negative effect on the sensory analysis (taste, odour, and overall acceptability)	[88]
Chicken	Breast meat	Rosemary (*R. officinalis* L.) powder + vitamin E	0.5 g/kg	-	14	Increase in oxidative stability	[89]
Chicken	Minced thigh meat	Pomegranate (*Punica granatum* L.) pomace extract (PPE) and pomegranate pomace (PP)	0.3 g/kg (PPE) 3 g/kg (PP)	45 mg GAE/g meat (TPC; PPE), 42 mg GAE/g meat (TPC; PP)	11	Inhibitory effect on lipid oxidation (decreased TBARS values); increase in radical scavenging capacity; decrease of n-6/n-3 ratio	[90]
Chicken	Breast muscle, fat, liver	Tea polyphenols	15 g/kg	-	-	Inhibitory effect on oxidative stress induced by corticosterone; reduction of abdominal fat content, plasma triglyceride concentration and liver weight	[91]
Laying hens	Eggs	Grape (*V. vinifera*) pomace flour	30 g/kg	-	30	Increase of laid eggs percentage; decrease of yolk and albumen pH; reduction of egg lipid oxidation; decrease in contents of caproic, butyric and margaric fatty acids	[92]
Laying hens	Eggs	Dried orange pulp (*Citrus sinensis*)	90 g/kg	-	28 (room temperature) 90 (4 °C)	Improvement of egg yolk oxidative stability; reduction in feed intake; negative influence on egg quality (lighter eggs, lower eggshell percentage, thickness, and strength, less orange yolk colour)	[93]
Pig	*Longissimus muscle* muscle	Fermented grape (*V. vinifera*) pomace product	30 g/kg	-	-	Decrease of SFA^3^ (palmitic, stearic, and arachidic acids) and increase of PUFA concentrations; inhibitory effect on lipid oxidation (decreased TBARS values); higher redness and yellowness values	[94]
Pig	Raw and cooked *Longissimus dorsi* steaks	Grape (*V. vinifera*) seed extract (GSE) and bearberry (BB)	0.7 g/kg (GSE) 0.7 g/kg (BB)	-	16 (raw meat) 28 (cooked meat)	No effect on oxidative stability and quality of raw and cooked meat	[95]
Pig	Loin meat mixture	Green tea (*Camellia sinensis*) by-products	20 g/kg	-	-	Inhibitory effect on lipid oxidation (decreased TBARS values)	[96]
Pig	*Longissimus dorsi* muscle, lard	Rosemary (*R. officinalis* L.) extract	1 g/kg	-	5	Increase in PUFA content; increase of oxidative stability (decreased TBARS values)	[97]
Pig	*Longissimus lumborum* muscle	Rosemary (*R. officinalis* L.)	1 g/kg	-	5	No influence on meat lipid content; improvement in meat fatty acid profile (increase in the content of PUFA of the n-3 and n-6 series); no influence on shelf-life duration	[98]
Pig	*Longissimus lumborum* muscle, “capocollo” (neck pork meat)	Rosemary (*R. officinalis* L.) aqueous leaf extract (R), oregano (*Origanum vulgare*) aqueous leaf extract (O)	2 g/kg (R) 2 g/kg (O) 1 + 1 g/kg (OR)	2.8 GAE/g dry meat (TPC; O) 3.2 GAE/g dry meat (TPC; R) 4.1 GAE/g dry meat (TPC; OR)	-	Increase in antioxidant activity (higher GSH-Px values (O diet), greater appreciation of meat obtained from animals fed a diet supplemented with rosemary	[99]
Pig	Loin, bacon, chops patties	Cranberry (*Vaccinium macrocarpon*) juice powder	150 g/kg	-	-	No influence in oxidative stability; decrease of redness and colour intensity values	[100]
Pig	*Longissimus dorsi*	Tea polyphenols (TP), Lucta sweetener, flax oil (FO)	0.4 g/kg diet (TP)	-	-	Increase of T-SOD and GSH-Px (prevention of possible oxidations in the muscle)	[101]
Rabbit	*Longissimus dorsi*, hind legs	Rosemary (*R. officinalis* L.) aqueous extract (R), oregano (*O. vulgare*) aqueous extract (O)	2 g/kg (R) 2 g/kg (O) 1 + 1 g/kg (RO)	-	-	Improvement of oxidative stability (lower TBARS content, lower oxidation degree); improvement of protein content	[102]
Rabbit	Meat from *Longissimus lumborum* muscle	Olive (*Olea europaea* L.) pomaces	50 g/kg	-	-	Inhibitory effect of lipid oxidation; increase in MUFA content and decrease in PUFA content; no influence on meat physical traits	[103]
Rabbit	Meat from *Longissimus dorsi* muscle	Spirulina (*Arthrospira platensis*; S), thyme (*Thymus vulgaris*; T)	50 g/kg (S) 30 g/kg (T)	-	9	Reduction of lipid oxidation (T); improvement of colour parameters (T); improvement in α-tocopherol and n-3 fatty acids content (T); reduction of drip loss (T)	[104]
Rabbit	Meat from *Longissimus thoracis and lumborum* muscle	Bilberry (*Vaccinium myrtillus* L.) pomace	150 g/kg	-	60	No influence on carcass characteristics, proximate composition, lipid oxidation and consumer acceptance of rabbit meat; improvement of fatty acids profile (increase of total n-3 PUFA)	[105]
Dairy cattle	Milk, caciotta cheese	Dried grape (*V. vinifera*) pomace	100 g/kg DM	-	28 (caciotta cheese)	No influence on milk composition, increase in the concentration of linoleic acid and trans-vaccenic acid (milk and caciotta cheese); increase in rumenic acid (caciotta cheese); darker colouring, harder consistency, less sweet taste (caciotta cheese)	[106]
Dairy cattle	Milk	Grape (*V. vinifera*) residue silage	100 g/kg DM	12.3 mg GAE/L (TPC) 1.3 mg QE/L (flavonoids)	-	No influence on milk yield, fat, lactose, crude protein; decrease of urea nitrogen and total solids; increase in PUFA concentration and PUFA/SFA ratio; no influence on the concentration of total polyphenols and flavonoids	[107]
Dairy cattle	Milk	Pelleted citrus pulp (PCP), soybean oil	30 g/kg of soybean oil and 90 g/kg PCP (SOCP-9); 30 g/kg of soybean oil and 180 g/kg PCP (SOCP-18)	27.2 μg GAE/mL (SOCP-9; TPC) 31.1 μg GAE/mL (SOCP-18; TPC) 0.8 μg QE/mL (SOCP-9; flavonoids); 0.8 μg QE/mL (SOCP-18; flavonoids)	-	Increase in polyphenols and flavonoids content and total ferric reducing antioxidant power; decrease in SFA content; increase in MUFA content	[108]
Dairy cattle	Milk	Ensiled *Moringa oleifera*	271 g/kg DM	-	-	Improvement of milk yield, increase in milk antioxidant capacity, decrease in somatic cell count content	[109]
Dairy cattle	Milk	Propolis (*Baccharis dracunculifolia*) based product (PBP), flaxseed oil	1.2 g/kg DM	13.35 mg GAE/L (TPC; PBP)	-	Increase of fatty acids trans9–18:1, cis9, trans11–18:2 concentrations, and total CLA content; increase of total polyphenols concentration; increase of reducing antioxidant power	[110]
Beef cattle	Meat from *Longissimus thoracis* (LT) and *semitendinosus* (ST) steaks	Plant extracts rich in polyphenols (PERP) + vitamin E	7 g/kg	-	12	Protective effect against lipid oxidation (lower MDA values; after 12 days aging under-vacuum)	[80]
Beef cattle	*M. longissimus dorsi* (LD) steaks, cooked LD slices	Tea catechins (TC), rosemary (*R. officinalis* L.) extracts (RE)	1 g/animal/day (TC) 1 g/animal/day (RE)	-	8	No significant influence on lipid stability and surface redness (LD steaks); no influence on sensory properties and lipid stability (cooked LD slices)	[111]
Beef cattle	Meat from M. *Longissimus lumborum*	Dried citrus pulp (DCP), dried grape (*V. vinifera*) pomace (DGP)	150 g/kg DM	32.5 g GAE/kg dry matter (TPC; DCP) 89.6 g GAE/kg DM (TPC; DGP) 12.7 g GAE/kg DM (total tannins; DCP) 50.7 g GAE/kg DM (total tannins; DGP)	9	Improvement of antioxidant activity (DGP > DCP); inhibitory effect on lipid oxidation (DGP < DCP); higher value of lightness in beef; improvement of beef shelf life	[112]
Beef cattle	*Longissimus dorsi* steaks	Olive cake	5 g/kg DM	-	9	Inhibitory effect of lipid oxidation; delay of colour deterioration and off-odour during storage; reduction of peroxide value; no influence on microbial counts	[113]
Beef cattle	Fresh and cooked beef from *Longissimus* *thoracis* muscle	Dried citrus pulp (DCP)	800 g/kg DM	0.92 mg GAE/g muscle (TPC)	14 (fresh beef) 6 (cooked beef)	No influence on TBARS concentration (fresh beef); reduction of lipid oxidation (40% DCP in cooked beef); no negative influence on texture characteristics and consumer acceptability	[114]
Goat	Milk, cheese	Distilled thyme (*T. zygis* subsp. gracilis) leaves (DTL), not-distilled thyme leaves (TL)	200 g/kg DM (DTL) 75 g/kg DM (TL)	400 mg GAE/kg of cheese (TPC; DTL) 450 mg GAE/kg of cheese (TPC; TL)	45 (cheese)	Improvement of milk quality (increase in PUFA content); inhibitory effect on lipid oxidation (TBARS values decreased; TL); improvement of sensory characteristics (DTL: best rind and taste)	[115]
Goat	Milk	*Acacia farnesiana* pods	300 g/kg DM	305.5 mg GAE/L of milk (TPC)	-	Improvement of antioxidant activity (scavenging capacity increased, higher Trolox equivalent value, higher ferric reducing antioxidant power value)	[116]
Goat	Meat from M. *longissimus dorsi* (LD), *gluteus medius* (GM) and *semimemberanosus* (SM) muscles	Tea catechins (TC) from green tea (*C. sinensis*) leaves	4 g/kg	-	-	Improvement of antioxidant activity (TBARS values decreased), improvement of meat colour stability; decrease of drip loss percentage (GM); decrease of intramuscular fat (SM)	[117]
Goat	Meat from M. *longissimus thoracis* and *lumborum* muscle	*M. oleifera* leaves	200 g/animal/day	1.62 ± 0.27 mg GAE /g of meat (T)	-	Improvement of antioxidant activity; increase in scavenging potential; increase of percentage of inhibition against lipid oxidation; increase in SOD activity	[118]
Goat	Meat from *Muscularis longissimus thoracis* and *lumborum* muscle	Olive mill wastewaters (OMWW) powder extract	0.0032 g/day	-	7	Increase in MUFA concentration, decrease in SFA concentration; inhibitory effect of lipid oxidation (reduction of MDA content); no significant influence on meat proximate composition; texture and colourimetric properties	[119]
Sheep	Pecorino cheese	Rosemary (*R. officinalis* L.) leaves	2.5%	2.14 mg GAE/g of cheese (TPC after 3 weeks from the start of the trial) 2.62 mg GAE/g of cheese (TPC after 5 weeks from the start of the trial) 3.60 mg GAE/g of cheese (TPC after 7 weeks from the start of the trial)	-	Increase in the total polyphenol concentration in milk and Pecorino cheese; increase in antioxidant activity of cheese; inhibitory effect on lipid oxidation; no influence on cheese composition; modification of overall cheese flavour (after 7 weeks from the start of the trial)	[120]
Sheep	Milk, Pecorino cheese	Fresh lemon (*Citrus limon*) pulp	2000 g/day	10.4 g GAE/kg DM (TPC)	-	Influence on milk fatty acid composition (improvement of vaccenic and rumenic acids); increase in milk protein and casein percentages; increase in antioxidant activity of cheese and total content of phenolic compounds	[121]
Sheep	Raw and cooked minced lamb meat	Rosemary (*R. officinalis* L.) extract	0.6 g/kg	-	21	Extension of raw meat shelf life (lipid oxidation and rancidity delayed), no influence on the shelf life duration of cooked meat	[122]
Sheep	Meat fillets from *Longissimus dorsi* muscle	Rosemary *(R. officinalis* L.) extract	0.006 g/kg	-	21	Inhibitory effect on meat lipid oxidation (MDA levels reduced) and rancidity, prevention of sensory deterioration	[123]
Sheep	Milk	Concentrated Pomegranate (*P. granatum* L.) peel extract	451 g/DM/kg	-	-	Increase in milk antioxidant capacity, improvement of milk yield and composition (higher phospholipid, fat, protein and lactose content)	[124]

CLA: conjugated linoleic acid; DM: dry matter; GAE: gallic acid equivalents; MDA: malondialdehyde; MUFA: monounsaturated fatty acids; PUFA: polyunsaturated fatty acids; QE: quercetin; TBARS: thiobarbituric acid reactive substances; TPC: total phenolic content.

**Table 3 animals-11-00401-t003:** Various sources used as natural antioxidants for meat and dairy products.

Product	Source	Dose	Storage Conditions	Concentration Product	Effect	Reference
Cooked chicken breast meat	Grape (*Vitis vinifera*) seed, green tea (*Camellia sinensis*) extracts	2.5 g/kg	4 °C for 12 days	-	Inhibitory effect on lipid oxidation; improved texture of meat; no influence on colour parameters and pH	[176]
Mechanically deboned poultry meat	Rosemary (*Rosmarinus officinalis* L.) extracts (dried extract (D), aqueous extract (WE), 40% ethanol extract (E40), 70% ethanol extract (E70), essential oil (EOS)	20 g/kg (D, WE, E40, E70) 2 g/kg (EOS)	Vacuum plastic bags at −18 °C for 4 months	-	Inhibitory effect on lipid oxidation and microbial growth (E70, EOS the most effective treatments)	[177]
Raw and cooked minced broiler meat	Lemon (*Citrus limonum*), orange (*C. sinensis*) and grapefruit (*C. paradise*) seed extracts	15 g/kg	Foil paper at 4 °C for 12 days	-	Inhibitory effect on lipid oxidation (in both raw and cooked meat samples); antioxidant effect more pronounced in raw meat samples	[178]
Raw chicken meat	Cloves *(Syzygium aromaticum,* SA), Chinese cinnamon (*Cinnamomum cassia,* CC), oregano, (*Origanum* *vulgare;* OV), and black mustard (*Brassica nigra,* BN)	10 g/kg	Meat aerobically packed in low-density polyethylene bags at 4 °C for 15 days	-	Inhibitory effect on lipid oxidation (combination of SA, CC and OV: lowest TBARS values and higher lightness, redness and yellowness values); reduction of microbial growth	[179]
Cooked chicken patties	Pomegranate (*Punica granatum*) rind powder (RP)	0.2 g equivalent RP phenolics/kg meat	Patties packaged in LDPE pouches at 4 °C for 15 days	441.00 ± 41.78 µg TAE/g (TPC)	Inhibitory effect on lipid oxidation (TBARS values remained low up to 15 days); reduction of lightness value (chicken patties became slightly darker) and yellowness value; no influence on sensory parameters	[180]
Chicken meatballs	Cinnamon (*Cinnamomum verum*) deodorized aqueous extract	0.2 g/kg	Meatballs covered with an oxygen semi-permeable PVC film and stored in dark at 8 ± 1 °C for 12 days	-	Inhibitory effect on lipid oxidation (decreased TBARS and POV values); no negative influence on sensory acceptability; decrease of lightness value; no influence on redness and yellowness values	[181]
Chicken breast meat	Thyme (*Thymus vulgaris*), lemon balm (*Melissa officinalis*) essential oils	5 g/kg	Meat packaged in sterile conical tubes under aseptic conditions at 4 °C for 21 days	-	Reduction of DPPH radical formation; reduction of lipid peroxidation and the deterioration of sarcoplasmic proteins; extension of the shelf life; reduction of total microbial counts	[182]
Rabbit meat patties	Fermented rooibos (*Aspalathus* *linearis*) extract	20 g/kg	Patties wrapped with PVC and stored at 4 ± 1 °C for 7 days	-	Inhibitory effect on lipid oxidation; protective action against protein degradation; reduced SFA content, no influence on PUFA content; reduced lightness value, increased redness and yellowness values; undesired sensory characteristics (>0.5% level inclusion)	[183]
Raw and cooked rabbit burgers	Ginger (*Zingiber officinale* Roscoe) powder	20 g/kg	Burgers overwrapped with polyethylene film and stored at 4 ± 0.5 °C for 7 days	-	Inhibitory effect on lipid oxidation; improvement of antioxidant capacity; increased total PUFA percentage	[184]
Fresh and stored rabbit burgers	Turmeric (*Curcuma longa* L.) powder	35 g/kg	Burgers overwrapped with polyethylene film and stored at 4 ± 0.5 °C for 7 days	-	Improvement of antioxidant capacity; no influence on lipid oxidation; reduced yellowness value; increased PUFA content	[185]
Raw ground pork meat	Mustard (*Brassica juncea*) leaf *kimchi* extracts	2 g /kg	Anaerobic polyethylene/Nylon film bags at 4 ± 1 °C for 14 days	-	Inhibitory effect on lipid oxidation; extension of shelf-life; reduction of total bacterial count	[186]
Raw ground pork meat	Curry (*Murraya koenigii L.*), mint leaves (*Mentha spicata*)	0.25 g/kg	LDPE at 4 ± 1 °C for 12 days	-	Inhibitory effect on lipid oxidation; reduced pH value at days 6, 9 and 12 of storage; reduced redness and yellowness values; influence on colour stability	[187]
Pork patties	Rosemary (*R. officinalis* L.) extract (RE), tea (TE), grape (*V. vinifera*) skin extract (GSE) and coffee (CE) extracts	0.2 g/kg (RE, TE, GSE) 0.05 g/kg (CE)	Patties vacuum packaged at 4.5 °C for 10 days	-	Inhibitory effect on lipid oxidation (antioxidative efficiency: RE > GSE > TE > CE > control); reduced hexanal values	[188]
Cooked pork patties	Grape (*V. vinifera*) seed (GS), water-soluble oregano (*O. vulgare*) extract (WS), oleoresin rosemary (*R. officinalis* L.) (OR) extracts	0.2 g/kg	Patties wrapped in PVC at 4 °C for 8 days	-	Inhibitory effect on lipid oxidation (GS, WS); no influence on TBARS values (OR); no influence on colour and pH	[189]
Cooked sausages	*Cinnamomum zeylanicum* essential oil	0.04 g/kg	4 °C for 30 days	-	Inhibitory effect on lipid oxidation (decreased TBARS and POV values); no influence on sensory characteristics	[190]
Cooked sausages	Nutmeg (*Myristica fragrans*) essential oil	0.020 g/kg	4 °C for 60 days	-	Inhibitory effect on lipid oxidation; reduction of the total number of aerobic mesophilic bacteria; improvement of aroma	[191]
Raw and cooked pork patties from *Longissimus dorsi* muscle	Grape seed *(V. vinifera)* extract (GSE), bearberry (BB)	1 g/kg	MAP (raw pork; 75% O_2_:25% CO_2_) at 4 °C for 12 days MAP (cooked pork; 75% O_2_:25% CO_2_) at 4 °C for 4 days	-	Inhibitory effect on lipid oxidation on day 9 and 12 of storage (GSE, BB; raw pork); inhibitory effect on TBARS formation after 2 and 4 days of storage (cooked pork); increase in redness value of cooked pork (GSE); no influence on sensory parameters	[192]
Raw pork patties	Grape (*V. vinifera* and *V. labrusca* hybrid) seed extract (GRA), green tea (*C. sinensis*) extract (TEA), chestnut (*Castanea sativa*) extract (CHE), seaweed (*Ulva lactuca* and *U. rigida*) extract (SEA)	1 g/kg	Patties packed in polystyrene trays sealed with polyethylene film (80% O_2_–20% CO_2_) at 2 °C for 20 days	-	Inhibitory effect on lipid and protein oxidation (antioxidative efficiency: GRA > TEA > SEA > CHE); decreased total viable count (TEA and GRA: lower *Pseudomonas* counts); TEA and GRA: lower psychotropic aerobic bacterial counts	[193]
Salami	Peanut (*Arachis hypogaea* L.) skin extract	1 g /kg product	Salami stored at 65% relative humidity and 15 °C for 42 days	-	Inhibitory effect on lipid oxidation; preservative effect of sensory properties	[194]
Pork patties	Black currant (*Ribes nigrum* L.) extract (BCE)	20 g/kg	Patties overwrapped in PVC at 4 °C for 9 days	-	Inhibitory effect on lipid oxidation (efficacy of BCE comparable with BHA); inhibition of protein oxidation (decreased carbonyl formation and increased sulfhydryl groups content); improved redness value (patties with BCE displayed purple colour)	[195]
Pork patties	Mugwort (*Artemisia princeps* Pamp.), rosemary (*R. officinalis* L.)	0.5 g /kg	Patties anaerobically packaged in PE/nylon film bags at 4 °C for 15 days	-	Inhibitory effect on lipid oxidation (lower primary and secondary products of lipid oxidation); decreased chroma values	[196]
Cooked beef patties	Grape (*V. vinifera*) seed (GS), water-soluble oregano (*O. vulgare*) extract (WS), oleoresin rosemary (*R. officinalis* L.) (OR) extracts	0.2 g /kg	Patties wrapped in PVC at 4 °C for 8 days	-	Inhibitory effect on lipid oxidation (GS, WS); no influence on TBARS values (OR); no influence on pH; reduced visual green discolouration	[189]
Fresh beef steaks	Oregano (*O. vulgare* L.) extract (OE)	40 g/kg (active film containing OE) 0.4 mL/kg meat (direct addition of OE)	Steaks packaged with active film at 1 °C for 28 days	-	Inhibitory effect on lipid oxidation; increase of beef display life from 14 to 23 days; negative influence on odour (unacceptable oregano smell)	[197]
Raw and cooked low-fat beef patties	Plum (*Prunus domestica*) puree	150 g/kg	Patties wrapped with polyethylene film at −18 °C for 45 days	-	Inhibitory effect on lipid oxidation; decreased lightness value, increased redness value, decreased yellowness value (raw samples); no influence on lightness value, lower redness value (cooked samples); increased juiciness and texture scores	[198]
Raw beef patties	Chamnamul (*Pimpinella brachycarpa*), fatsia (*Aralia elata*) extracts	5 g/kg	Patties placed in Whirl-Pak bags at 4 °C for 12 days	-	Inhibitory effect on lipid oxidation; decreased redness value; reduction of final microbial load	[199]
Raw and cooked beef patties	Dog rose (*Rosa canina* L.) extract	50 g/kg	Patties stored in individual oxygen-permeable polyethylene bags at −18 °C for 20 weeks	-	Inhibitory effect on lipid and protein oxidation; higher moisture losses; increased hardness in patties subjected to frozen storage	[200]
Fresh beef patties	Lyophilized and powdered *Gentiana lutea* root	2 g/kg	Patties packaged in MAP (20 O_2_:80 CO_2_) at 4 ± 2 °C for 10 days; Patties packaged in MAP (80 O_2_:20 CO_2_) at 4 °C ± 2 °C for 10 days	-	Inhibitory effect on lipid oxidation; no influence on pH and microbial counts; no influence on sensory attributes (appearance and taste)	[201]
Irradiated ground beef	Marjoram (*O. majorana*), rosemary (*R. officinalis* L.), sage (*Salvia officinalis*)	0.4 g/kg	5 °C	-	Inhibitory effect on lipid oxidation; improved colour; decreased irradiation odour (increased overall acceptability)	[202]
Raw and cooked ground fresh goat meat	Green tea (*C. sinensis*) extract, grape (*V. vinifera*) seed extract	6 g/kg	Meat wrapped with oxygen-permeable PVC at 5 °C for 9 days	-	Inhibitory effect on lipid oxidation; reduction of redness value	[203]
Goat meat nuggets	Broccoli (*Brassica oleracea*) powder extract	20 g/kg	Nuggets aerobically packaged at 4 °C	0.16 ± 0.01 mg GAE/g (TPC)	Inhibitory effect on lipid oxidation; increased meat phenolic content; decreased pH; decreased yellowness and chroma values; no influence on sensory attributes	[204]
Cooked goat meat patties	Kinnow (*Citrus reticulate*) rind powder (KRP), pomegranate (*Punica granatum*) rind powder (PRP) and pomegranate (*P. granatum*) seed powder (PSP)	10 mL	Patties anaerobically packaged in LDPE bags at 4 °C for 12 days	900 µg TAE/g (TPC; KRP) 1200 µg TAE/g (TPC; PRP) 500 µg TAE/g (TPC; PSP)	Inhibitory effect on lipid oxidation; (antioxidative efficiency: PRP > PSP > KRP); increased lightness value (KRP), reduced lightness value (PRP); reduced redness values (PRP, PSP); decreased yellowness value (PRP); no influence on sensory evaluation for colour and appearance; reduced pH value (KRP < PSP < PRP)	[205]
Goat meat patties	Chrysanthemum *morifolium* flower extract	2 g/kg	Patties labelled at 4 °C for 9 days	-	Inhibitory effect on lipid and protein oxidation; no influence on organoleptic properties (appearance, flavour, texture, juiciness, and overall acceptability)	[206]
Fermented goat meat sausages	Rosemary (*R. officinalis* L.) powder extract	0.5 g/kg	Meat vacuum packaged at 30 °C for 90 days	-	Inhibitory effect on lipid oxidation; reduced loss of redness, prevention of off-flavour formation	[207]
Ground goat meat, goat meat nuggets	Pomegranate (*P. granatum*) peel extract	10 g/kg	Meat vacuum-packed at 4 °C for 9 days; Nuggets vacuum-packed at 4 °C for 25 days		Inhibitory effect on lipid oxidation; no influence on lightness value; reduced colour difference; slow growth of aerobic plate counts	[208]
Ground sheep meat	Sumac (*Rhus coriaria* L.) water extract, barberry (*Berberis vulgaris* L.) water extract	30 g/kg	Meat stored under aerobic conditions into polyethylene bags at 4 °C for 9 days	-	Inhibitory effect on lipid oxidation; inhibitory effect of microbial growth; improvement in odour score	[209]
Lamb patties	Olive oil waste extract	0.4 g GAE/kg muscle	Patties packaged in MAP (70% O_2_/30% CO_2_) at 4 °C for 9 days	-	Inhibitory effect on lipid and protein oxidation; lower fish odour and flavour, higher odd odour and flavour; acceptable meat after 6 days of storage	[210]
Sheep meat nuggets	Litchi (*Litchi chinensis* Sonn.) pericarp extract	15 g/kg%	Nuggets aerobically packed in LDPE pouches at 4 °C	0.17 ± 0.01 mg GAE/g (TPC)	Inhibitory effect on lipid oxidation; no influence on sensory attributes	[211]
Lamb patties (cooked and refrigerated-stored)	Hop (*Humulus lupulus* L.) infusion and powder	2 g/kg	Raw: 4 °C for 7 days; −18 °C for 90 days; Cooked: 4 °C for 3 days	-	Inhibitory effect on lipid and protein oxidation; modification in flavour acceptance (decreased acceptance of freshly prepared patties)	[212]
Sheep patties	Peanut (*A. hypogaea* L.) skin extract	1 g/kg	Patties packaged in MAP (80% O_2_—20% CO_2_) at 2 ± 1 °C for 20 days	-	Inhibitory effect on lipid and protein oxidation; reduction of redness loss; improvement of sensory attributes	[213]
Lamb burgers	Rosemary (*R. officinalis* L.), thyme (*T. vulgaris*) powder extracts	1 g/kg	Burgers covered with PVC at 2 °C for 6 days	-	Inhibitory effect on lipid oxidation; no influence on *Enterobacteriaceae* count	[214]
Lamb patties	Rosemary (*R. officinalis* L.), ginger (*Zingiber officinal*) extracts	0.5 g/ kg (rosemary) 5 g/kg (ginger)	Patties wrapped with polyethylene film at −18 °C for 5 months	-	Inhibitory effect on lipid oxidation; improved overall acceptability, flavour and colour; reduction of microbial growth	[215]
Ultra-filtrated soft cheese derived from buffalo’s skim milk retentate	Rosemary (*R. officinalis* L.) extract	50 g/kg	Cheeses stored at 6 °C for 30 days	615 mg GAE/100 mg (TPC)	Improvement of radical scavenging activity and ferric antioxidant power; improvement of sensory evaluation total scores	[216]
Stirred yoghurt	Pomegranate (*P. granatum*) peel extract (before and after inoculation with the starter)	350 g/kg	6 ± 2 °C	8.23 ± 1.54 mg GAE/g (TPC before inoculation) 7.61 ± 1.00 mg GAE/g (TPC after inoculation) 3.59 ± 0.78 µg RE/g (total flavonoid content before inoculation) 2.99 ± 0.01 µg RE/g (total flavonoid content after inoculation)	Improvement of antioxidant activity (higher total phenol content and total flavonoid content); no influence on sensory properties (flavour, body and texture, appearance)	[217]
Yoghurt	Neem *(Azadirachta indica)* leaves powder	-	Yoghurt stored at 4 °C for 28 days	74.9 ± 6.2 µg GAE/mL	Improvement of antioxidant activity and total phenolic content; no influence on organoleptic properties	[218]
Full-fat cheese	Green tea (*Camellia sinensis*) extract	1 g/kg	Cheese stored at 8 ± 2 °C for 90 days	900 GAE equivalent/100 g cheese (TPC)	Improvement of antioxidant activity and phenolic content; no influence on cheese composition (moisture, protein and fat)	[219]
Yoghurt from cow, buffalo and goat milk	Ginger (*Zingiber officinale*) extract and beetroot (*Beta vulgaris*) extract	20 g/kg	Yoghurt stored at room temperature	-	Improvement of antioxidant activity (goat milk yoghurt > cow milk yoghurt > buffalo milk yoghurt)	[220]
Cheddar-type cheese	Green tea (*C. sinensis*) extract	2 g/kg	Cheese packed under vacuum and stored at 4 °C for 29 days	3.91 g/kg	Improvement of antiradical activity; no influence on total proteins, fat, salt, micellar calcium content of cheese; moisture decreased; influence on colour (loss of lightness, increase in redness and yellowness); increase in hardness and flavour intensity; firmness increased	[221]
Yoghurt	Hazelnut (*Corylus avellana* L.) skins var. ("Tonda Gentile Trilobata" (TGT), ’San Giovanni’ and ‘Georgia’)	60 g/kg	Yoghurt stored at 4 °C for 21 days	19.43 ± 1.84 µg GAE/g DM (TPC; TGT) 17.86 ± 0.80 µg GAE/g DM (TPC; San Giovanni) 18.48 ± 0.25 µg GAE/g DM (TPC; Georgia)	Improvement of antioxidant capacity; decreased consumer acceptance	[222]
Yoghurt	Green tea (*C. sinensis*) extract (GTE), black tea extract (BTE) and white tea extract (WTE)	20 g/kg	Yoghurt stored at 4 °C for 21 days	3220.15 ± 37.80 µg GAE/mL (TPC; GTE) 2811.26 ± 44.74 µg GAE/mL (TPC; BTE) 2504.59 ± 24.48 µg GAE/mL (TPC; WTE)	Improvement of antioxidant capacity (GTE > WTE > BTE)	[223]
Cow and camel milk	Soybean (*Glycine max* L.)	-	Yoghurt stored at 4 °C for 21 days	43.17 ± 1.2 µg GAE/g (TPC; cow milk) 91.76 ± 1.8 µg GAE/g (TPC; camel milk)	Improvement of antioxidant activity (in camel milk); increased viability of *Lactobacillus* spp.	[224]
Yoghurt	Chamomile (*Matricaria recutita* L.) and fennel (*Foeniculum vulgare* Mill.) decoctions	0.4 g/kg	Yoghurt stored at 4 °C for 14 days	-	Improvement of antioxidant activity (chamomile > fennel); no influence on pH and colour parameters	[225]
Ghee	Coriander *(Coriander sativum* L.) extract	5 g/kg	80 ± 1 °C for 21 days	-	Improvement of radical scavenging activity; inhibition of peroxide and conjugated dienes formation; improvement of antioxidant activity	[226]
Paneer	Peels of pomegranate (*P. granatum*), orange (*Citrus* sinensis) and lemon (*C. limonum)*	30 g/kg	Paneer stored for 8 days	-	Improvement of antioxidant activity (prevention of peroxide formation: pomegranate extract > lemon extract > orange extract)	[227]
Milk and yoghurt	Red ginseng (*Panax ginseng*) extract	20 g/kg	Milk and yoghurt stored at 4 °C	41.1 ± 0.9 mg of AE/100 g (TPC; yoghurt) 38.3 ± 0.8 mg of GAE/100 g (TPC; milk)	Improvement of antioxidant activity	[228]

BHA: butylated hydroxyanisole; DM: dry matter; DPPH: 1,1-diphenyl-2-picrylhydrazyl radical scavenging activity; FFA: Free fatty acids; GAE: gallic acid equivalent; LDPE: low-density polyethylene; MAP: modified atmosphere packaging; POV: peroxide value; PUFA: polyunsaturated fatty acids; PVC: polyvinyl chloride; SFA: saturated fatty acids; TAE: tannic acid equivalent; TBARS: thiobarbituric acid reactive substances; TPC: total phenolic content.

## Data Availability

All data referred to in the manuscript are already published.

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
