# Peer review of "Dietary Polyphenol Supplementation in Food Producing Animals: Effects on the Quality of Derived Products"

_animals, 2021, doi:10.3390/ani11020401_

Round 1

Reviewer 1 Report

This review summarizes relative findings by summarizing the literature of recent years about polyphenols, including their classification and structure, natural distribution, bioavailability in food-producing animals, their antioxidant function and application in food products, which offers an overview of the effects of polyphenols both supplemented to the diet of monogastric and ruminants and added directly to meat and dairy products with their special effects. This is a beneficial and meaningful article, and I think this article can be published in the journal after some errors have been corrected.

Major comments:

  1. At the beginning of the part of “Introduction”, why do the authors refer to lipid oxidation firstly and specifically while lipid oxidation seems not the key point of this article.
  2. Why is not the secondary heading of “non-flavonoids” in Figure 1 listed?
  3. It is better to add some references to verify the “3.8 Lignans”.
  4. From line 202 to 208, is there literature supporting phenolic compounds’ pathways of digestion, absorption, and subsequent metabolism?
  5. The antioxidant effects of polyphenolic compounds on lipids, proteins, and even vitamins are mentioned in section 5, but there is no literature to support these roles.
  6. The plant sources listed in Table 2 are best mentioned in Part 3, so that it is possible to know which substances of polyphenolic compounds play a specific role, such as the pomegranate.
  7. There are too few summaries of Table 2.
  8. The “simple summary” and “abstract” sections are best reworked according to the main points of the article.

Specific comments:

  1. The sentence in Line 12-13 lacks a subject.
  2. Line 48, it is better to use “ether…or”.
  3. The usage of “either-or” needs to be checked, it should be used to join two words or phrases of the same nature
  4. Line261-262, this sentence exists a grammar mistake.
  5. Line 372, the word, “content”, is better to be deleted.
  6. In lines 616 and Line 619, the sentence needs to be rewrite and some mistakes should be checked, such as the punctuation.
  7. Some abbreviations need to be specified, such as in part 7.

Author Response

Response to Reviewer 1 Comments

This review summarizes relative findings by summarizing the literature of recent years about polyphenols, including their classification and structure, natural distribution, bioavailability in food-producing animals, their antioxidant function and application in food products, which offers an overview of the effects of polyphenols both supplemented to the diet of monogastric and ruminants and added directly to meat and dairy products with their special effects. This is a beneficial and meaningful article, and I think this article can be published in the journal after some errors have been corrected.

Thanks for your effort and time spent on our manuscript; we really appreciate your helpful comments. We have made point to point revision according to your constructive suggestions.

Major comments:

  1. At the beginning of the part of “Introduction”, why do the authors refer to lipid oxidation firstly and specifically while lipid oxidation seems not the key point of this article.

Thanks for your comment; we modified a little the beginning of introduction and we deleted the sentence related to lipid oxidation. Anyway, we think that polyphenols having antioxidant properties are connected to lipid oxidation concern. Lines 38-40.

  1. Why is not the secondary heading of “non-flavonoids” in Figure 1 listed?

Thank you so much for your carefully attention. The Figure 1 has now been modified and the term “non-flavonoids” has been added. Line 79.

  1. It is better to add some references to verify the “3.8 Lignans”.

A reference (number 32) has been added, thank you for the suggestion. Line 162.

  1. From line 202 to 208, is there literature supporting phenolic compounds’ pathways of digestion, absorption, and subsequent metabolism?

A reference (number 37) has been added, thanks for the suggestion. Line 204.

  1. The antioxidant effects of polyphenolic compounds on lipids, proteins, and even vitamins are mentioned in section 5, but there is no literature to support these roles.

Thank you for this suggestion. Some references supporting the effects of polyphenols on lipids, proteins and vitamins have been added. Lines 312, 321, 330, 331, and 334.

  1. The plant sources listed in Table 2 are best mentioned in Part 3, so that it is possible to know which substances of polyphenolic compounds play a specific role, such as the pomegranate.

We are not really sure what the reviewer would like to see. We mentioned Table 2 in Section 3, since Table 2 includes specific role referred to substances of polyphenolic compounds as described in Part 3. We hope we have answered satisfactorily. Lines 185-186.

  1. There are too few summaries of Table 2.

Table 2 reports results not discussed in the text; therefore, in Table 2 there are additional findings on the effects of polyphenols supplemented to the diet of monogastric and ruminants on the physicochemical and sensorial properties of the final product (meat and dairy products).

  1. The “simple summary” and “abstract” sections are best reworked according to the main points of the article.

Thank you for the suggestion, the Simple Summary and Abstract have now been changed to better describe the main points of the review. Lines 10-19 (Simple Summary) and Lines 20-33 (Abstract).

Specific comments:

  1. The sentence in Line 12-13 lacks a subject.

The reviewer is correct; sorry for the mistake. The subject has now been added. Line 13.

  1. Line 48, it is better to use “ether…or”.

Thank you for the suggestion, the sentence has now been modified as follows: “Recently, the interest of food processing industries in the use of natural antioxidants rather than synthetic counterparts was increased, for either low environmental impact or economic reasons.” Lines 44-46.

  1. The usage of “either-or” needs to be checked, it should be used to join two words or phrases of the same nature.

Thank you for the suggestion. The use of “either-or” has been expanded. Lines 411-412, lines 540-541.

  1. Line 261-262, this sentence exists a grammar mistake.

Thanks for your advice. The sentence has now been rewritten as follows: “However, a direct interaction between some specific phenolic compounds (such as hydrolysable tannins) and methanogen microbes was suggested”. (Lines 260-262).

  1. Line 372, the word, “content”, is better to be deleted.

We agree with reviewer and deleted the word. Line 399.

  1. In lines 616 and Line 619, the sentence needs to be rewrite and some mistakes should be checked, such as the punctuation.

The reviewer is correct. The sentence has now been corrected as follows: “The results obtained suggest that grape seed extract could represent a technologically viable alternative to stabilize lipid oxidation in beef meat as well”. Lines 646-648.

A dot was missing, it has now been added. Line 650.

  1. Some abbreviations need to be specified, such as in part 7.

Thanks for your carefully attention. Some abbreviations have now been specified. TBARS (Line 388-389), LD (Line 403), EPA, DHA, ALA, (Lines 405-406), MDA (Line 408), GSE (Line 633).

Reviewer 2 Report

This review article evaluates the use of polyphenols in animal agriculture both at the whole animal level and within the products produced by these animals.

This review appears to be fairly comprehensive. Unlike other reviews which discuss this topic, this review focuses on the potential role of these factors in food science applications as well as livestock production. There is a great deal of information in this review with regards to the role of polyphenols in meat/dairy products. 

 However, additional work needs to be done in order to prepare this review for publication. 

Major Issues:

1. Readability is impacted by long wordy sentences as well as oddly worded sentences that need additional clarification. Furthermore, references need to be provided more frequently and in some cases additional references are needed to support statements made. Tables need some reformatting. For example to words need to be on all one line if possible. Please see the following examples by line of the issues mentioned. As a whole, the paper should be revised to fix similar issues.

Line #39: "Evolving towards"; oddly worded

Animal fat health: is this referring to metabolic activity of the adipose tissue. Please clarify

Line #49: Are polyphenols considered to be GRAS compounds? This is a designation provided by the FDA in the US. Are there similar designations used globally for these compounds?

Line #51: How would these boost nutritional value. Need a reference here.

Line #54-55: Repetitive; doesn’t fit here

Line #202: Metabolism or digestion?

Line 204-207:Long sentence

Line #212: May need to use the word animal here instead of organism as organism can mean bacteria too.

Line #217: On the other hand is an informal term that should not be used.

2. Additional information needs to be added in to clarify statements made or to account for missing information that needs to be discussed. Please see below for examples.

The role of these polyphenols in livestock needs to be expanded further and additional references need to be provided. How do these compounds impact ADG, feed intakes, growth rates etc in the different species? Furthermore clarifications on some of the statements made needs to be done.

Line 81: Mention Tannis and what appears to be a negative impact of this compound on production traits in livestock. A single reference is provided which appears to be a review article focusing on humans. Livestock specific references need to be used.

Do the bacteria or processes of digestion alter the chemical structure of these compounds? It is mentioned in the review but this is an area that could be expanded upon. It would also be helpful to add in information on how these compounds are metabolized in animals (Line 233-234).

Line #266 it is mentioned how growth can be regulated by these compounds. How is this done? This is a very bold statement and additional information needs to be provided with appropriate references.

Line #318: Additional information needs to be provided to outline cost analyses/studies that have been done to establish the cost effectiveness.

Author Response

Response to Reviewer 2 Comments

This review article evaluates the use of polyphenols in animal agriculture both at the whole animal level and within the products produced by these animals.

This review appears to be fairly comprehensive. Unlike other reviews which discuss this topic, this review focuses on the potential role of these factors in food science applications as well as livestock production. There is a great deal of information in this review with regards to the role of polyphenols in meat/dairy products. 

However, additional work needs to be done in order to prepare this review for publication. 

We are grateful for your valuable comments and suggestions, which were very useful to improve our paper. We have carefully revised the manuscript and highlighted all changes. Please find our response on the questions asked below.

Major Issues:

  1. Readability is impacted by long wordy sentences as well as oddly worded sentences that need additional clarification. Furthermore, references need to be provided more frequently and in some cases additional references are needed to support statements made. Tables need some reformatting. For example to words need to be on all one line if possible. Please see the following examples by line of the issues mentioned. As a whole, the paper should be revised to fix similar issues.

There was an overlap in the reviewers' comments regarding the request to add reference to support some statements. We accordingly provided them, we shortened long wordy sentences in order to improve the readability and Tables were managed and reformatted.

Line #39: "Evolving towards"; oddly worded

Animal fat health: is this referring to metabolic activity of the adipose tissue. Please clarify

We agree with reviewer and modified the whole sentence as follows: “There is a growing interest in producing healthier animal products with a higher ratio of polyunsaturated (PUFA) to saturated fatty acids by modulation of animal’s diet. This nutritional strategy has been associated with an increase in lipoperoxidation.” Lines 38-40.

Line #49: Are polyphenols considered to be GRAS compounds? This is a designation provided by the FDA in the US. Are there similar designations used globally for these compounds.

To the best of our knowledge no equivalent globally designation exist. Anyway, a designation for safe similar to GRAS could be the one provided by EFSA for feed and food, limited to grape seed in relation to risk issue; for food the health claim is limited to olive oil polyphenols.

European Food Safety Authority (EFSA) Panel on Additives and Products or Substances used in Animal Feed (FEEDAP) has established the safety and efficacy of dry grape extract used as feed flavoring for all animals and categories except dogs at a maximum level of 100 mg/kg complete feed. The panel emphasized no safety concerns for consumers at this level in feeds (EFSA FEEDAP Panel (EFSA Panel on Additives and Products or Substances used in Animal Feed), 2016).

In the Journal of AOAC International Vol. 102, No. 5, 2019 1373, the following conclusion is reported

At present, the use of health claims related to polyphenols in the European market is limited only to olive oil polyphenols. All other formulations so far proposed relating to other foods have been rejected by the European Authority, being judged as lacking substantial scientific evidence. We did not add the above information.

Line #51: How would these boost nutritional value. Need a reference here.

Thank you for the suggestion. Two references have been added (references number 2 and 3). Line 50.

Line #54-55: Repetitive; doesn’t fit here

Thanks for your suggestion. The sentence has been removed. Line 52.

Line #202: Metabolism or digestion?

Metabolism is correct. Line 198.

Line 204-207: Long sentence

Thank you for the suggestion. The sentence has been divided into two shorter sentences as follows: “The route of phenolic absorption can either be via the stomach and small intestine or possibly absorbed by the colon after chemical modification by the colonic microbiota. The microbiota present in the colon allows polyphenols to be absorbed into the bloodstream and subsequently to be excreted either in the urine or via the bile.” Lines 200-204.

Line #212: May need to use the word animal here instead of organism as organism can mean bacteria too.

Thank you for the suggestion, the term “organism” has been replaced by “animal”. Line 209.

Line #217: On the other hand is an informal term that should not be used.

Thank you for this suggestion. The term “on the other hand” has been deleted. Line 213.

  1. Additional information needs to be added in to clarify statements made or to account for missing information that needs to be discussed. Please see below for examples.

The role of these polyphenols in livestock needs to be expanded further and additional references need to be provided. How do these compounds impact ADG, feed intakes, growth rates etc in the different species? Furthermore clarifications on some of the statements made needs to be done.

Thank you for the suggestion. Additional information with relative references has now been added in the text. Lines 282-305.

Line 81: Mention Tannis and what appears to be a negative impact of this compound on production traits in livestock. A single reference is provided which appears to be a review article focusing on humans. Livestock specific references need to be used.

A reference focused on the use of tannins in livestock animals replaced the reference focusing on humans (reference number 9). Consequently, the sentence has been modified as follows: “They have been reported to be responsible for decreases in feed intake, growth rate, feed efficiency, net metabolizable energy, and protein digestibility in livestock animals.” Lines 76-78.

Do the bacteria or processes of digestion alter the chemical structure of these compounds? It is mentioned in the review but this is an area that could be expanded upon. It would also be helpful to add in information on how these compounds are metabolized in animals (Line 233-234).

Yes, bacteria or processes of digestion modify the chemical structure as explained from Line 198 to Line 222. We added the term “transformed” missing in the first submission (Line 209). We expanded this aspect adding a schematic illustration as new figure (Figure 3) according also to request of reviewer 3 who asked more figures in order to catch the reader’s attention and stress the most important aspects.

Line #266 it is mentioned how growth can be regulated by these compounds. How is this done? This is a very bold statement and additional information needs to be provided with appropriate references.

Thank you for the suggestion. Some references on mechanism that links productive performance with antioxidant action due to polyphenols (oxidative stress-inflammation-effects performance) have been added in the text to support this statement. Please see from Line 278 to Line 305.

Line #318: Additional information needs to be provided to outline cost analyses/studies that have been done to establish the cost effectiveness.

To the best of our knowledge no specific studies focused on analysis on costs have been performed. However, the increase in number of publications (Figure 4 in the revised text, Figure 3 in the first submission) and positive results (Table 2 and Table 3) seem to establish the cost effectiveness and the great interest in the use of polyphenols in the animal’s diet both as feed and as commercial extracts.

We agree with Brenes et al., 2016 (reference number 37), who reported that “It was highlighted that polyphenol-rich grape by-products could partially replace costly vitamin E in monogastric diets by finding the right proportion to combine them”.

In the last part of the abstract (Lines 31-33) we highlighted this concept adding the following sentence: “However, the inclusion of agro-industrial by-products rich in polyphenols, in animal feed, represents an innovative and alternative source of antioxidants as well as being useful in reducing environmental and economic impact”.

We are aware that for a definitive judgment on polyphenols, their concentration (feed/food) should be considered and quantification of simple compounds, more expensive than analysis on total phenolic content, currently represents an important and growing field of research.

We hope that this answer fulfills your concern.

Reviewer 3 Report

The submitted manuscript is a long and detailed review on dietary polyphenol supplementation in food producing animals and products originating from those animals. I like the tables that summarize in a nice way the previously published studies. My detailed comments are presented below.

Lines 11 and 40: What do you mean by „animal fat health”?

Line 16: Exactly 8000 or more than 8000?

Figure 1, first of all please improve its quality. The numbers next to lignans are not needed. Please unify the way the structures are presented. Further, actually none of the presented structures is strictly a polyphenol as none of them has more than one phenolic group. Finally, why some names start with capital letter (i.e. Quercetin) and some not (i.e. riodicytol). Concluding, I like the idea behind this figure (presenting the classes of polyphenols with examples of  chosen compounds) but its quality must be improved.

Line 76: Cinnamic acid, not cinnamic acids.

Line 104, for Vitis vinifera please use italics.

Line 107: Not alcohol group but hydroxyl group.

Line 111, what is a 3-hydroxygroup ?

Lines 113-114, why Coenzyme with capital C?

Line 183: “polymerswith” and “Themonomeric”, spaces are needed. Actually, the whole manuscript should be read carefully in order to remove many editorial mistakes.

Line 226: “bodily tissues” should be rewritten

Line 256, again some names using capital letters, this is really not needed.

Please move Figure 3 right after it is mentioned for the first time, maybe in line 313?

Figure 3, please improve its quality and dpi. Please copy it from excel as a windows metafile, do not make a screenshot.

Line 361: Please define TBARS.

Line 521, change 0.1 kg to 100 g.

Line 619: A dot is missing.

The total number of Figures (only 3) is very low for a such long manuscript. Please think about at least one-two more to 1)catch the reader’s attention 2)stress the most important aspects.

Author Response

Response to Reviewer 3 Comments

The submitted manuscript is a long and detailed review on dietary polyphenol supplementation in food producing animals and products originating from those animals. I like the tables that summarize in a nice way the previously published studies. My detailed comments are presented below.

We thank the positive appraisal from our reviewer on our work and the manuscript itself as well as her/his accurate comments on the text.

Lines 11 and 40: What do you mean by „animal fat health”?

The term “animal fat health” has been removed, since we changed both the Simple Summary (Lines 10-19) and the sentence according also to reviewer 1 (Lines 38-40).

Line 16: Exactly 8000 or more than 8000?

The Simple Summary has been modified according to reviewer 1, and the term 8000 has been removed. Lines 10-19.

Figure 1, first of all please improve its quality. The numbers next to lignans are not needed. Please unify the way the structures are presented. Further, actually none of the presented structures is strictly a polyphenol as none of them has more than one phenolic group. Finally, why some names start with capital letter (i.e. Quercetin) and some not (i.e. riodicytol). Concluding, I like the idea behind this figure (presenting the classes of polyphenols with examples of chosen compounds) but its quality must be improved.

Thank you for the suggestions. The quality of Figure 1 has now been improved, and some changes have been made, also in agreement with the reviewer 1. Line 79.

Line 76: Cinnamic acid, not cinnamic acids.

Done as suggested. Line 73.

Line 104, for Vitis vinifera please use italics.

Thanks for your attention. We corrected. Line 100.

Line 107: Not alcohol group but hydroxyl group.

Thanks for your attention. We corrected. Line 103.

Line 111, what is a 3-hydroxygroup?

The term has now been modified as follows: “hydroxyl group at the 3-position”. Line 107.

Lines 113-114, why Coenzyme with capital C?

Thanks for your attention, the word has been corrected using lowercase letter. Lines 109-110.

Line 183: “polymerswith” and “Themonomeric”, spaces are needed. Actually, the whole manuscript should be read carefully in order to remove many editorial mistakes.

Spaces have now been added and the whole manuscript has been checked to avoid any other such mistakes. Thanks for your carefully attention. Line 179.

Line 226: “bodily tissues” should be rewritten

Corrected; the term “bodily” has been replaced by “body”. Line 226.

Line 256, again some names using capital letters, this is really not needed.

Done as suggested (rumenic acid and vaccenic acid). Line 256.

Please move Figure 3 right after it is mentioned for the first time, maybe in line 313?

Thank you for the suggestion, Figure 3 (now renamed Figure 4) has been moved above. Line 341.

Figure 3, please improve its quality and dpi. Please copy it from excel as a windows metafile, do not make a screenshot.

The quality of Figure 3 (now renamed Figure 4) has been improved. Thank you for the suggestion. Line 341.

Line 361: Please define TBARS.

Sorry for the forgetfulness, the acronym TBARS has now been defined (Lines 388-389). It has also been defined under the Table 2 (Line 346) and Table 3 (Line 593).

Line 521, change 0.1 kg to 100 g.

Thank you for this correction. Line 550.

Line 619: A dot is missing.

A dot has been added. Line 650.

The total number of Figures (only 3) is very low for a such long manuscript. Please think about at least one-two more to 1) catch the reader’s attention 2) stress the most important aspects.

The reviewer is correct. Thank you for this suggestion. A new figure (Figure 3) has been added to illustrate the adsorption and metabolism of plant polyphenols in monogastric farm animals. Line 224.

Round 2

Reviewer 3 Report

The Authors have significantly improved their manuscript, I find it suitable for publication.

Author Response

Response to Reviewer 3 Comments

The Authors have significantly improved their manuscript, I find it suitable for publication.

Thanks for the time spent on our manuscript. We are grateful for your valuable comment and for giving us the opportunity to improve our work.
